# Examining the flavor descriptors of e-cigarettes, heated tobacco products, and nicotine pouches in the Philippines: Regulatory challenges and opportunities

**Samantha J. Ackary**⬥°*, **Patrik James D.L. Cabrera**°, **Alen Josef A. Santiago**⬥°*, **Gianna Gayle H. Amul**⬥°

Ateneo Policy Center, School of Government, Ateneo de Manila University, Quezon City, Metro Manila, Philippines

° These authors contributed equally to this work.

* sackary@ateneo.edu (SA); ajsantiago@ateneo.edu (AS)

## Abstract

In the Philippines, the regulation of e-cigarettes, heated tobacco products (HTPs), and nicotine pouches under Republic Act 11900 or the Vape Law is crucial as their use continues to rise. As e-cigarettes are rapidly gaining popularity due to their social appeal, perceived novelty, ever-changing flavors, and sensations that are attractive to consumers, HTPs and nicotine pouches are continuously developing as well. This study aims to characterize e-cigarette, HTP, and nicotine pouch flavor descriptors in the Philippines and to assess the regulatory implications of existing policies regulating these products. We adapted the Tobacco Pack Surveillance System (TPackSS) data collection protocol developed by the Institute for Global Tobacco Control and employed both inductive and deductive thematic analyses to categorize flavor descriptors from 278 e-cigarette, HTP, and nicotine pouch products. We identified 11 categories of flavor descriptors: colors (18.08%), fruits (15.07%), codes or acronym-like descriptors (13.70%), concept descriptors (12.05%), menthol (10.96%), beverages (9.86%), sweet (8.22%), miscellaneous (5.48%), pop culture (3.56%), tobacco (2.47%), and nuts (0.55%). We also examined the flavor imagery, marketing tactics, and promotional strategies on these products. To effectively regulate e-cigarettes, HTPs, and nicotine pouches, we recommend streamlining their governing policies. Establishing a complete flavor ban on appealing flavors, descriptors, and imagery is crucial to curbing use. Additionally, increasing taxation and implementing plain packaging can reduce the marketing appeal of these products, thereby enhancing prevention and control efforts.

## Introduction

The Philippines suffers from 112,000 deaths every year from tobacco-related diseases [1]. However, while tobacco use dropped from 29.7% in 2009 to 19.5% in 2021 [2], the Philippines is experiencing a rising epidemic of e-cigarette use. The prevalence of e-cigarettes use amongst adults rose from 2.8% in 2015 to 5.7% in 2021, with current users growing from 0.8% to 2.1%

**Data availability statement:** The data supporting the findings of this study are included as supplementary material to this manuscript.

**Funding:** This work was supported by the help of a grant from the Department of Health (DOH), Republic of the Philippines (to SA, PC, AS, and GA). The contents of this document are the sole responsibility of the authors and can under no circumstances be regarded as reflecting the positions of Ateneo de Manila University (ADMU) or DOH. The views expressed herein are those of the authors and do not necessarily reflect the views of ADMU or the DOH. The funders had no role in study design, data collection and analysis, decision to publish, or preparation of the manuscript.

**Competing interests:** The authors have declared that no competing interests exist.

[3,4]. E-cigarette use is significantly higher among the youth where 25% of Filipino students aged 13–15 have tried e-cigarettes and 14% or one in seven currently use them [5]. This is alarming, considering that the current legal age for e-cigarette use in the Philippines is set at 18 years old [6]. As e-cigarette use continues to climb, the health implications, particularly the risk of e-cigarette or vaping-use associated lung injury (EVALI), pose an emerging public health danger. As of 2020, 2,807 hospitalized EVALI cases and deaths have been reported to the U.S. Centers for Disease Control [7]. Likewise, Southeast Asian countries including Thailand and Malaysia have reported increasing cases of EVALI [8]. Since May 2024, the Philippines has recorded six cases of EVALI, with one of the cases being the death of a 22-year old male [8,9]. The rising prevalence of e-cigarette use, together with emerging health risks such as EVALI, emphasizes the urgent need for effective policy measures to mitigate this growing public health threat.

The e-cigarette market in Southeast Asia is expected to grow by 29% by 2023, largely in Indonesia and the Philippines [10]. Southeast Asian youth are vulnerable targets of the e-cigarette market, which boasts a wide diversity of flavors, trendy designs, and point-of-sale promotions as key tactics in attracting the youth [11]. Flavors are the most common motivation for using e-cigarettes [12]. In particular, fruit and candy flavors are preferred and often lead to higher satisfaction and perceived addiction [13]. Additionally, flavored e-cigarettes are popular among young people because the flavors add appeal and mask the taste of nicotine [13,14].

Emerging risks from tobacco such as heated tobacco products (HTPs) and novel tobacco products like nicotine pouches pose a similar threat to the youth and adolescents. HTPs can attract adolescents by offering attractive flavors and sleek designs [15]. Nicotine pouch flavors are continuously developing, which could contribute to the initiation and sustained usage of nicotine pouches for both youth and adults [16]. Hence, these findings underscore the need for effective regulations on flavored e-cigarettes, HTPs, and nicotine pouches [15,16]. In the Philippines, e-cigarettes, HTPs, and nicotine pouches, both with and without nicotine, are currently regulated by the 2022 Republic Act (RA) 11900 or the Vape Law, which purportedly seeks to protect smokers, safeguard minors, generate revenue, and prevent illicit trade of these products [6].

As of 2021, 37 countries have banned e-cigarettes [17]. Out of 88 countries that allow the sale and regulation of e-cigarettes, at least 13 have restricted flavors other than tobacco in e-cigarettes. However, it is unclear whether these restrictions cover both flavorings and descriptors [18]. The U.S. Food and Drug Administration (FDA) restricted flavors to tobacco and menthol in 2020, leading some youths to switch to exempt products [19]. Finland banned characterizing flavors in 2016, but enforcement faces challenges due to the wide diversity of e-cigarette products, limited resources for advancing tobacco control, and retailer reluctance to complying with stricter flavor regulations [20]. While initiating and enforcing flavor regulations can be difficult, there is a recognized need for it, especially in Asian-Pacific countries that have no e-cigarette regulations [21].

In 2022, the Philippine FDA began regulating e-cigarette flavors through FDA Administrative Order (A.O.) 2022–1069, which prohibits vapor products with flavorings or additives other than plain tobacco or menthol [22]. However, the passage of the Vape Bill in 2022, now known as RA 11900 or the Vape Law, lifted this flavor ban through its repealing clause, which nullifies previous regulations on e-cigarettes. Instead of prohibiting certain flavorings and additives, the Vape Law only prohibits "unduly appealing" flavor descriptors, or those that "include a reference to a fruit, candy brand, dessert, or cartoon character" [6]. This offered the opportunity for e-cigarette retailers to continue the distribution and sale of flavored e-cigarettes, provided that the flavor descriptors do not appeal to minors. In addition to this,

the Vape Law overturned several other e-cigarette regulations such as reducing the minimum age of access from 21 years old to 18 years old; allowing the online sale and advertising of vaporized nicotine and non-nicotine products and novel tobacco products; and transferring regulatory jurisdiction from the Department of Health's (DOH) FDA to the Department of Trade and Industry (DTI) [23]. As a result, the Vape Law faced opposition by the Philippines' DOH, Department of Education, and various medical associations and civil society groups for reversing wins in local e-cigarette control [24].

Evidence shows that the global e-cigarette market is rapidly growing, consisting of diverse products and flavors aggressively marketed toward children and adolescents [25]. For this reason, in December 2023 the World Health Organization declared urgent action needed to protect children and the youth from the uptake of e-cigarettes, recommending interventions such as flavor bans and increased taxation [26]. The DOH of the Philippines recognizes that stronger implementation is still required to further prevent and reduce e-cigarette use [27], especially in light of tobacco industry interference in policy-making for e-cigarette control [23]. Effective implementation must include evidence-based tobacco control measures, of which flavor bans must be included to protect the youth from initiating and sustaining e-cigarette use [23].

While the online marketing of e-cigarettes and heated tobacco products has been previously examined [28], no study has yet categorized the range of flavored e-cigarettes, HTPs, and novel tobacco products at various points of sale in the Philippines. This study aims to describe the flavor landscape of e-cigarettes, HTPs, and nicotine pouches from various points of sale by examining and categorizing their flavor descriptors. Moreover, this study aims to examine flavor imagery that characterize these flavor descriptors, along with other strategies such as promotional and marketing tactics featured on these products. This study also outlines key implementation gaps in the Vape Law and the implications of such gaps in regulating flavored e-cigarettes, HTPs, and nicotine pouches.

## Methods

### Design

This study adapted the research design and methodology of the Tobacco Pack Surveillance System (TPackSS) and its 2015 Field Collection Protocol, originally developed by the Institute for Global Tobacco Control (IGTC) at the Johns Hopkins Bloomberg School of Public Health [29]. We made modifications to account for data unavailability (e.g., income level of *barangays*) in selecting the sites for data collection.

The data collection period took place from August 2023 to September 2023. Following the TPackSS protocol, our team gathered unique e-cigarette, HTP, and nicotine pouch products from the largest city in each of the three island groups of the Philippines: Antipolo City (Balanced Luzon), Cebu City (Visayas), and Davao City (Mindanao). We also collected products in Quezon City of the National Capital Region due to its deemed economic and geographical significance. We chose 12 *barangays* or neighborhoods from each city based on specific characteristics including population density, geographical location, and economic importance (S1 Table). We determined economic importance by identifying key economic or commercial hubs in the neighborhood, given the absence of income class data at the neighborhood level. These three characteristics were chosen to ensure logistical convenience, representativeness of the sample, and a diversity of products to access. Given that data on the socioeconomic classification was unavailable from online sources and local government authorities, we used these characteristics as proxies for socioeconomic classification. The data regarding population density, economic importance, and geographical location were based on both the Philippine Statistics Authority census results and the national tax allotment data by the Department of Budget and Management.

The inclusion criteria of this study encompasses all products regulated by the Vape Law, which includes vapor products, vapor product systems, refills, heated tobacco products, and novel tobacco products (specifically nicotine pouches) with flavor descriptors. All the terms used for the product types in this manuscript reflect the terminology stipulated under the Vape Law (S2 Table).

Within each neighborhood, our team sampled six main types of vendors: *sari-sari* stores (or neighborhood stores), convenience stores, mall kiosks, street vendors, supermarkets, and department stores. Starting from a commercial hub in each neighborhood, our team walked for five minutes until we encountered one of the six specified vendor types. Next, the protocol calls for the purchase of all unique products in the vendor. Unique products, according to the TPackSS protocol, are products with distinct designs or features, including variations in size, brand presentation, colors, cellophane wrapping, and inclusion of promotional items. However, in some cases, our team was unable to purchase all unique products from each vendor we visited. This is due to a limited budget and the extensive diversity particularly of e-cigarette products. In these cases, we prioritized buying at least one sample per every available brand of e-cigarette product at the vendor to minimize biases in product selection. In each neighborhood, we exhausted all available vendors in the area.

In a neighborhood, we considered data collection as complete if we collected as many unique products as we could that we had not already purchased from a previous vendor. In subsequent neighborhoods, our team purchased one of each unique product not already obtained from previous vendors. Upon completing purchases for the day, we assigned each product a code, photographed them following the TPackSS guidelines, and placed them in individual ziplock bags. We organized all photos into Google Drive folders for efficient documentation. No human participants were involved in the conduct of this study; hence, no ethics clearance was deemed necessary by the university institution.

## Coding

Following the collection of sample packs, our team adapted the TPackSS codebook to assign codes to each pack [30]. The codebook covers various packaging elements, including the dimensions of the package, the shape of the package, brand descriptors, and flavor descriptors. Flavor descriptors are defined as terms or phrases that convey or suggest characteristics and/or qualities in flavors of food, beverages, and others. Given that the initial TPackSS codebook was created for tobacco products, our team added additional elements to code for e-cigarettes, including product type based on definitions stipulated in the Vape Law (e.g., e-liquid refill, a device, etc.), the presence of nicotine, nicotine level, e-liquid capacity, and number of puffs. We also noted down marketing tactics such as the presence of merchandise and promotional strategies. Flavor imagery and device shape were recorded as ad-hoc notes while compliance of all products with the Vape Law were systematically assessed.

We conducted two rounds of coding. Two trained researchers conducted the first round of coding. The objective of the first round of coding was to assess if the sample meets the criteria for inclusion in the study, to familiarize the researcher with the samples collected, and to compile an initial list of codes. Each researcher coded all the packages included in the study. The objective of the second round of coding was to assess and validate the initial list of codes generated in the first round, with the help of a third researcher.

## Sample

Our team collected a total of 313 vapor products, vapor product devices, vapor product systems, heated tobacco products, heated tobacco product devices, heated tobacco product systems,

refills, and novel tobacco products in which 278 products fit our inclusion criteria. Our inclusion criteria covers vapor products, vapor product systems, refills, heated, tobacco products, and novel tobacco products (nicotine pouches) with flavor descriptors. Vapor products comprise pods or cartridges for vapor product devices. Vapor product systems are products and devices assembled into one unit, such as disposable vapes. Refills are e-liquid stored in cylindrical containers. In this manuscript, the use of the term "e-cigarettes" will collectively include all vapor products, vapor product devices, vapor product systems, and refills. HTPs include sticks or consumables used to load HTP devices. HTP systems include the product and device intended to be used together. Lastly, novel tobacco products included two nicotine pouches.

## Analysis

We conducted a thematic analysis using a combination of inductive and deductive approaches. We drew deductive codes from a 2016 study conducted by the Institute for Global Tobacco Control, which identified flavor descriptor categories such as menthol, beverages, fruit or citrus, and concept descriptors, which are "terms that imply that some type of flavor, sensation, taste, or aroma awaits the consumer" [31]. Simultaneously, we used an inductive approach to identify new flavor descriptor categories not covered in the 2016 study by IGTC. This entailed reviewing the data for recurring terms and themes, leading to new categories such as pop culture and codes or acronym-like descriptors.

We first listed all flavor descriptors individually then categorized them into distinct codes and sub-codes. For example, menthol flavors were broken down into sub-codes like mint and ice. The deductive codes provided structure, while the inductive process allowed us to capture additional nuances in flavor descriptors. Finally, we synthesized these codes into overarching themes, ensuring that both predefined and newly identified descriptors were included.

## Results

### General Information

Our team collected 313 products from 36 stores across a total of 49 *barangays* or neighborhoods in Quezon City, Antipolo City, Cebu City, and Davao City (S1 Table). Among these 313 products, 278 (88.89%) featured flavor descriptors (Table 1) and fall under the inclusion criteria of this study. The majority of products with flavor descriptors are vapor product systems (37.77%), followed by refills (32.37%) and  vapor products (25.18%).

We purchased half of the products from mall kiosks (50.71%) followed by street vendors (38.57%) and convenience stores (38.57%) (S3 Table). Given that malls are ubiquitous in highly urbanized cities in the Philippines, malls are regarded as key commercial hubs and the key place of purchase for any consumer product. As such, malls became a common data

Table 1.  **Product type distribution for products with and without flavor descriptors.**

| Product Type | Count with Flavor Descriptors | Count without flavor descriptors |
|---|---|---|
| Vapor Product | 70 (25.18%) | 0 (0.00%) |
| Vapor Product Device | 1 (0.36%) | 26 (74.29%) |
| Vapor Product System | 105 (37.77%) | 0 (0.00%) |
| Heated Tobacco Product | 10 (3.60%) | 6 (17.14%) |
| Heated Tobacco Product Device | 0 (0.00%) | 3 (8.57%) |
| Refill | 90 (32.37%) | 0 (0.00%) |
| Nicotine pouches | 2 (0.72%) | 0 (0.00%) |
| **Total** | **278** | **35** |

collection starting point for our team. We found that mall kiosks offered a large and diverse range of products compared to other types of stores.

We have identified 46 manufacturers across all samples of products (S4 Table); however, about 32.14% of products did not specify their manufacturer. Shenzhen RELX Technology Co., Ltd. (9.64%) and Shenzhen Xuewe Technology Co., Ltd. (9.29%) were the top manufacturers (S4 Table). Our analysis of the 280 products revealed that a significant majority (56.52%) originated from China, indicating the country's dominance in the e-cigarette market (Table 2). The diverse range of flavor descriptors as reflected in Table 3 suggests a strong emphasis on innovation and consumer preference, with products from the Philippines (26.09%) potentially reflecting local taste influences. Additionally, the limited representation of producers from countries like Italy and Poland points to emerging e-cigarette markets that may introduce unique flavor profiles in the Philippines.

## Flavor descriptors

Table 3 outlines the breakdown of the flavor descriptor categories along with examples of flavor descriptors. We classified 363 flavor descriptors into seven categories: colors (18.08%), fruits (15.07%), codes (13.70%), concept descriptors (12.05%), menthol (10.96%), beverages (9.86%), sweet (8.22%), miscellaneous (5.48%), pop culture (3.56%), tobacco (2.47%), and nuts (0.55%). Some descriptors fit multiple categories.

*Color* descriptors can take different forms, such as simple color names (e.g., emerald), descriptive phrases (e.g., purple snow), or more imaginative terms (e.g., tropical gold) (Fig 1). *Fruit* flavor descriptor come as standalone fruits, fruit mixes, or fruit-flavored food or beverages. Examples include "grapefruit," "kiwi guava," or "mango milkshake" (Fig 2). *Code* flavor descriptors are presented as acronym-like words that code certain flavors. For example, "STRW" is code for strawberry and "BBALL" is code for butterball (Fig 3). We decoded the product flavors using codebooks provided by retailers, which listed flavor descriptor codes alongside their actual flavors. We also cross-checked for inconsistencies between the product packaging and the device, as there were instances where coded flavor descriptors appeared on the packaging but not on the device itself (e.g., "STRW on the packaging but "strawberry" on the device). However, identifying the flavors was not always possible, especially in cases where codebooks were unavailable.

*Concept descriptors* are "terms that imply that some type of flavor, sensation, taste, or aroma awaits the consumer" [31]. Examples of these include "jungle fusion," "mellow melody," and "dream swirl" (Fig 4). *Menthol* flavor descriptors are divided into three subcategories: *menthol*, *mint*, and *ice* (Fig 5). Examples include "menthol xtra," "blue mint," or "peach

**Table 2. Manufacturer distribution by country of products with flavor descriptors.**

| Manufacturer Country of Origin | Count | Percentage |
|---|---|---|
| China | 26 | 56.52% |
| Philippines | 12 | 26.09% |
| Unknown | 2 | 4.35% |
| Japan | 1 | 2.17% |
| Italy | 1 | 2.17% |
| Hong Kong | 1 | 2.17% |
| USA | 1 | 2.17% |
| Poland | 1 | 2.17% |
| Indonesia | 1 | 2.17% |
| Total manufacturers | 46 | 100.00% |

**Table 3. Flavor descriptor categories breakdown.**

| Flavor Descriptor | Count | Percentage | Subcategory | Examples |
|---|---|---|---|---|
| Colors | 66 | 18.08% | | Tangy Purple<br>Tropical Gold<br>Amber Selection<br>Crimson Combo<br>Luscious Green |
| Fruits | 55 | 15.07% | | Strawberry Apple Banana<br>Passion Fruit Sunrise<br>Lychee Ice<br>Kiwi Guava<br>Sour Apple |
| Codes | 50 | 13.70% | | MCM (Mocha Cookie Milkshake)<br>BBall (butterball)<br>SIC<br>GinPom<br>CADOAVO |
| Concept Descriptors | 44 | 12.05% | | Jungle Fusion<br>Mellow Melody<br>Dreamy Swirl<br>Frozen oasis, chilled dopamine, tropical mist, bubble burst<br>Rich Roast |
| Menthol | 40 | 10.96% | Menthol<br><br>Mint<br><br><br>Ice | Menthol Xtra<br>Menthol Plus<br>Cool Fresh<br>Spearmint Intense<br>Minty Spears<br>Ice Fresh<br>Ice Mango |
| Beverages | 36 | 9.86% | Alcoholic beverage<br><br><br>Coffee<br><br><br>Milk<br><br><br>Soda<br><br><br>Others | Pina Colada<br>Lemon Mojito<br>Grapefruit Lime Cocktail<br>Latte Coffee<br>Choco Macchiato<br>Mocha Latte<br>Oat Milk<br>Cantaloupe Milk<br>Banana Milk Ice<br>Ice Cola<br>Orange Soda<br>Ribena<br>Pink Lemonade<br>Ginger Tea<br>Red bull sorbet |
| Dessert/Sweet/Candy | 30 | 8.22% | Dessert<br><br><br>Candy<br><br><br>Branded sweets | Mango Cheesecake<br>Ube brazo de mercedes<br>Special Halo-halo<br>Watermelon Candy<br>Grapes bubblegum<br>Pastillas supreme<br>Melona<br>Twinkies RY4<br>Dynamite Candy |
| Miscellaneous | 20 | 5.48% | | Bazooka[+]<br>Lucid Dream<br>Granny's Fave<br>Garden's Heart<br>Venus Fantasy |

*(Continued)*

**Table 3.** (Continued)

| Flavor Descriptor | Count | Percentage | Subcategory | Examples |
|---|---|---|---|---|
| Pop Culture | 13 | 3.56% | Slang words<br><br>People | Petmalu<br>Fucc Boi<br>Pink OG<br>Harlequin<br>Black Mamba<br>Mia Khalifa |
| Tobacco | 9 | 2.47% | | Virginia Tobacco<br>Brightleaf Tobacco<br>Classic Tobacco<br>Richman's Tobacco |
| Nuts | 2 | 0.55% | | Almond RY4<br>Hazelnut Brown |
| Total | 363 | 100.00% | | |

ice." *Beverage* flavor descriptors include subcategories such as *alcoholic beverages* (e.g., "pina colada"), *coffee* (e.g., "latte coffee"), *milk* (e.g., "oat milk"), and *soda* (e.g., "ice cola") (Fig 6).

*Sweet* flavor descriptors include three subcategories, namely *dessert* (e.g., "ube brazo de mercedes," a traditional Filipino dessert), *sweets* (e.g., "watermelon candy"), and *branded sweets* (e.g., "Twinkies RY4") (Fig 7). RY4 is a common blend of tobacco, vanilla, and caramel [32]. Given that RY4 was a recurring attachment to flavor descriptors in more than one category, we reviewed the official websites of e-cigarette companies and retailers to confirm the flavor descriptor RY4. *Miscellaneous* flavor descriptors denote objects, places, or people that do not necessarily connote a flavor, sensation, taste, or experience. Examples include "bazooka+," "lucid dream," and "granny's fave" (Fig 8). *Pop culture* flavor descriptors are subcategorized into *slang words* (e.g., "fucc boi" or "*petmalu*," an anagram of the Filipino word *malupit* which means "cool") and *people*, whether real or fictional (e.g., "Harlequin" or "Black Mamba," the nickname of the late NBA basketball player Kobe Bryant) (Fig 9). *Tobacco* flavor descriptors ranged from "Virginia tobacco," "brightleaf tobacco," and "richman's tobacco" (Fig 10). Lastly, only two *nut* flavor descriptors were discovered: "Almond RY4" and "hazelnut brown" (Fig 11).

## Flavor imagery

Our team observed that visual cues play an integral role in the packaging of flavored e-cigarettes, HTPs, and nicotine pouches. In particular, images are used to connote the flavor of the product, while iconographies are used to indicate coolness, sweetness, or nicotine levels of the product.

## Imagery

Expressive images are often used on the packaging of products in this study. Pictures of objects directly related to the flavor or flavor descriptor of the product are common. For example, ice-flavored products may feature pictures of ice cubes splashed in water, or watermelon-flavored products may feature pictures of watermelons. Images may be displayed as realistic photos of such objects, or as cartoon-like renderings (S1 Fig). Some brands offered unique flavor series (e.g., "Yakult series," a famous fermented milk drink in Asia) with related imagery (e.g., realistic images of the Yakult drinks). Disposable e-cigarettes used vibrant colors and textures, with some brands including a picture of the e-cigarette device for visualization.

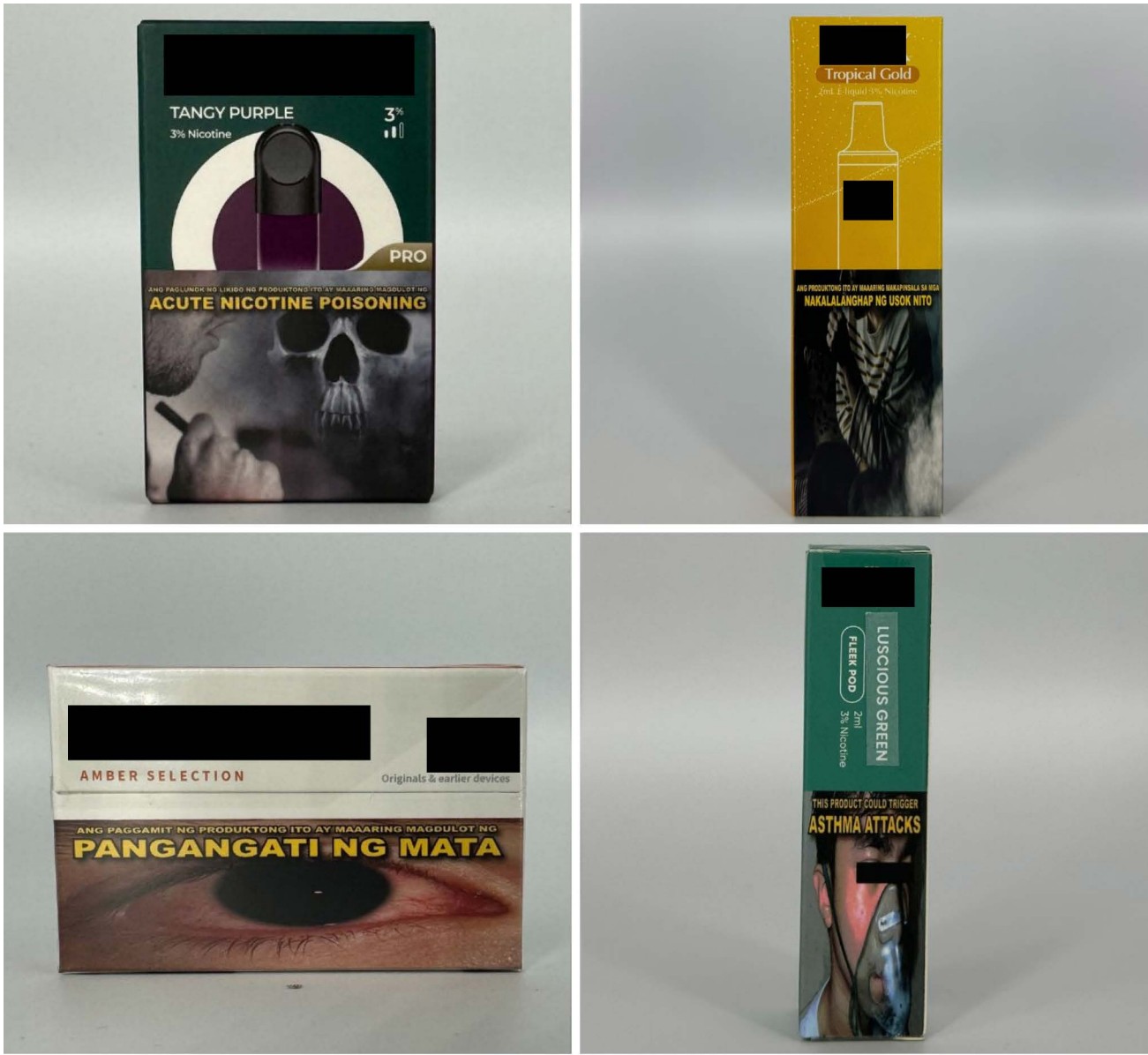

**Fig 1. Color flavor descriptors.**

## Iconography

Iconography indicated coolness, sweetness, or nicotine intensity levels of the product, usually represented as a scale of intensity (S2 Fig). In most cases, these intensity levels are clearly and pre-indicated on the packaging. In other cases, the scale is illustrated as check boxes that are meant to be manually checked off, in which we found some check boxes left blank. For example, some e-liquid refills with check boxes for nicotine levels of the product were left blank, leaving us unable to identify the nicotine dosage of the product. We also found other cases where the nicotine level can be checked as either "low," "mid," or "high"; however, there was no indication of the actual dosage amount for each level.

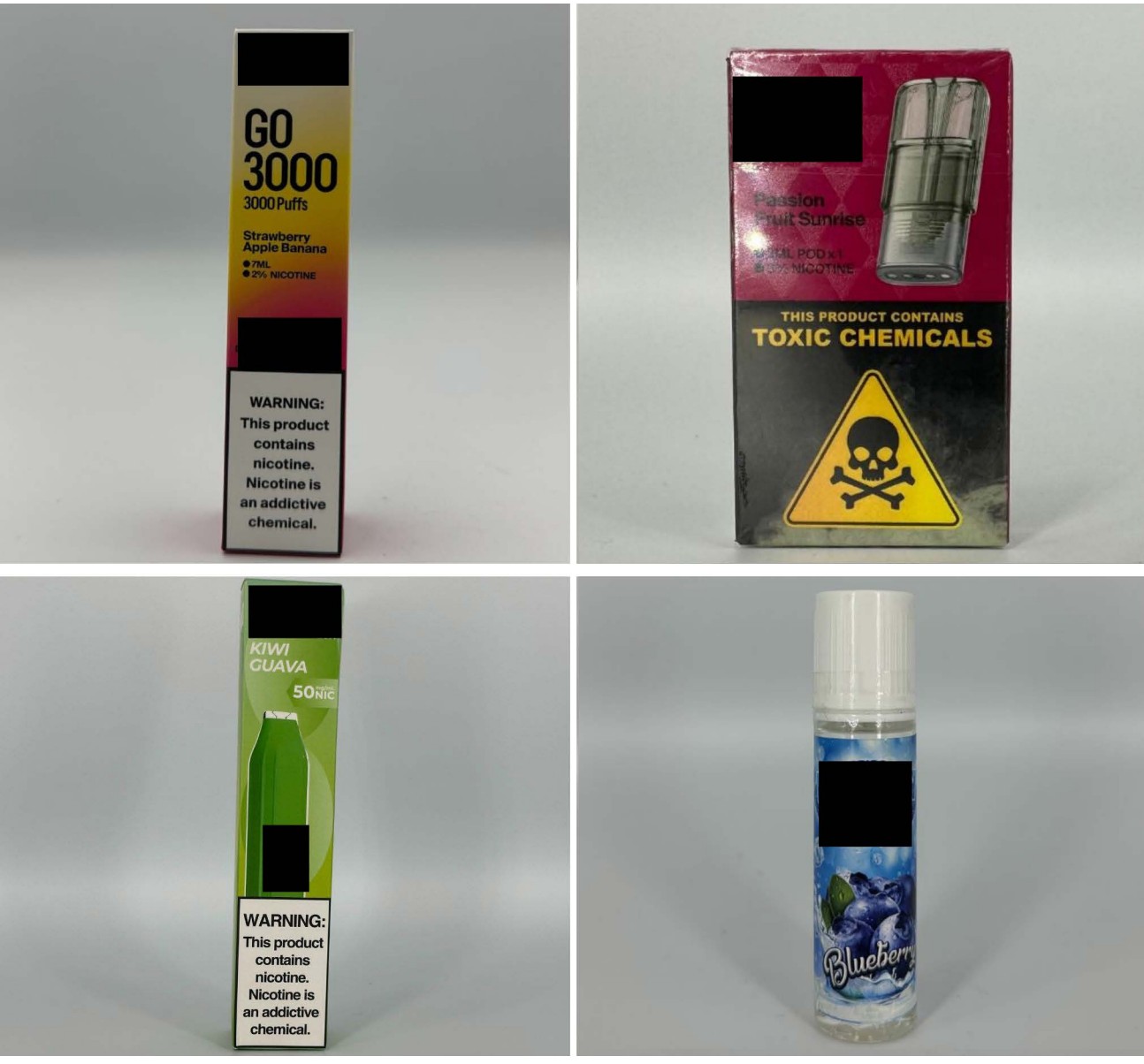

**Fig 2. Fruit flavor descriptors.**

## Marketing strategies

We examined marketing strategies by noting down factors ranging from unique device shapes and mechanisms, merchandise, and social media promotion.

## Unique device shapes and mechanisms

E-cigarette devices are commonly marketed as products that look like food or beverage items. We found devices that were shaped as milk cartons, energy drinks, and juice boxes (S3 Fig). Additionally, we collected devices that have unique mechanisms, including having the ability to shift between four flavors in one device by twisting the top of the device.

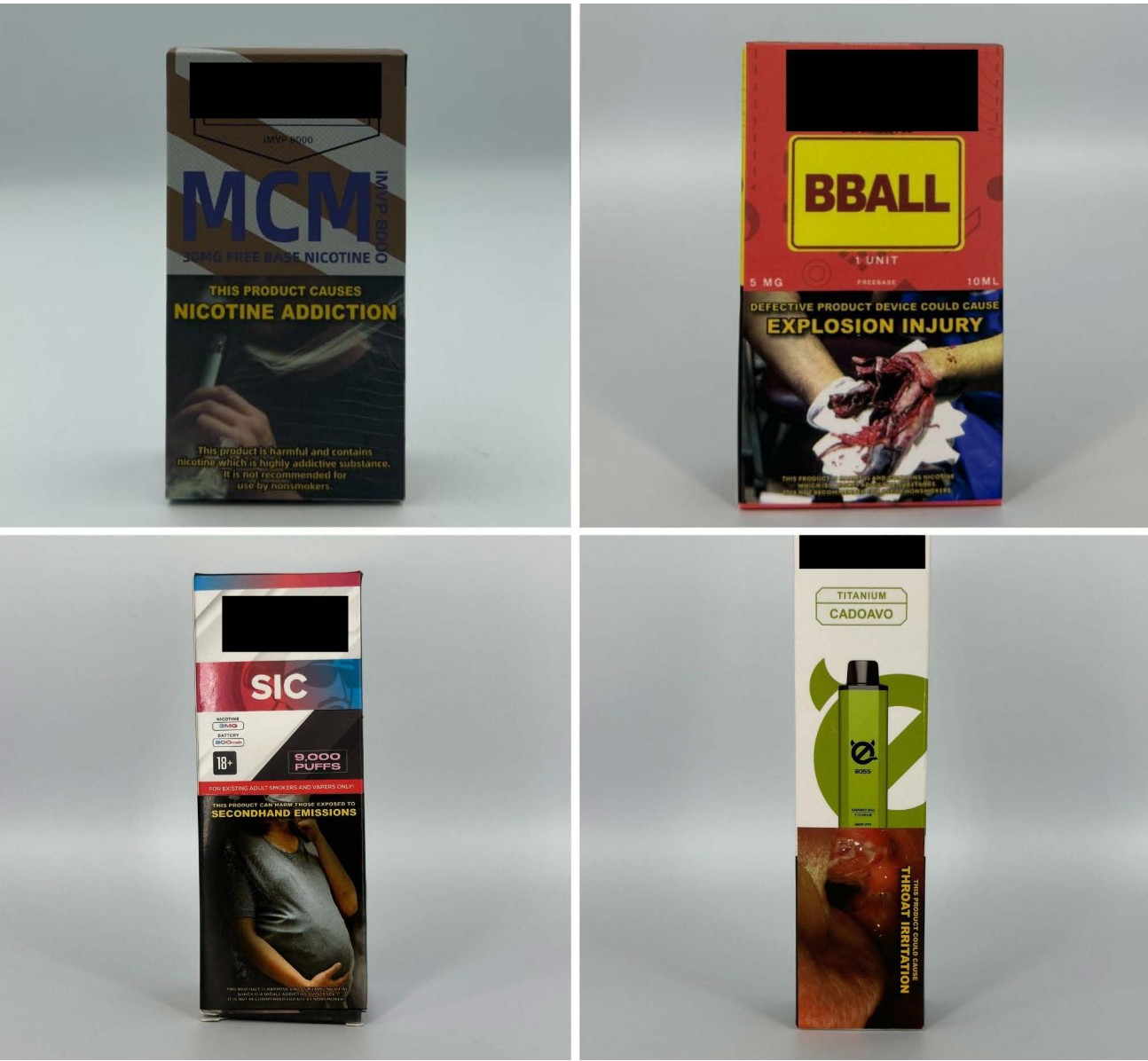

**Fig 3. Code flavor descriptors.**

### Merchandise and brand stretching

The inclusion of merchandise is common, either as joint inclusions in the product box or as a freebie. The practice of brand stretching is frequent, where e-cigarette brand names and logos are used on non-e-cigarette products such as lanyards and mobile phone chargers (S4 Fig) [33]. In one instance, we received a counterfeit Adidas pouch as a freebie.

### Social media promotion

Plugging social media accounts of the e-cigarette company is another common marketing tactic which involves presenting the social media handle and its corresponding platforms on

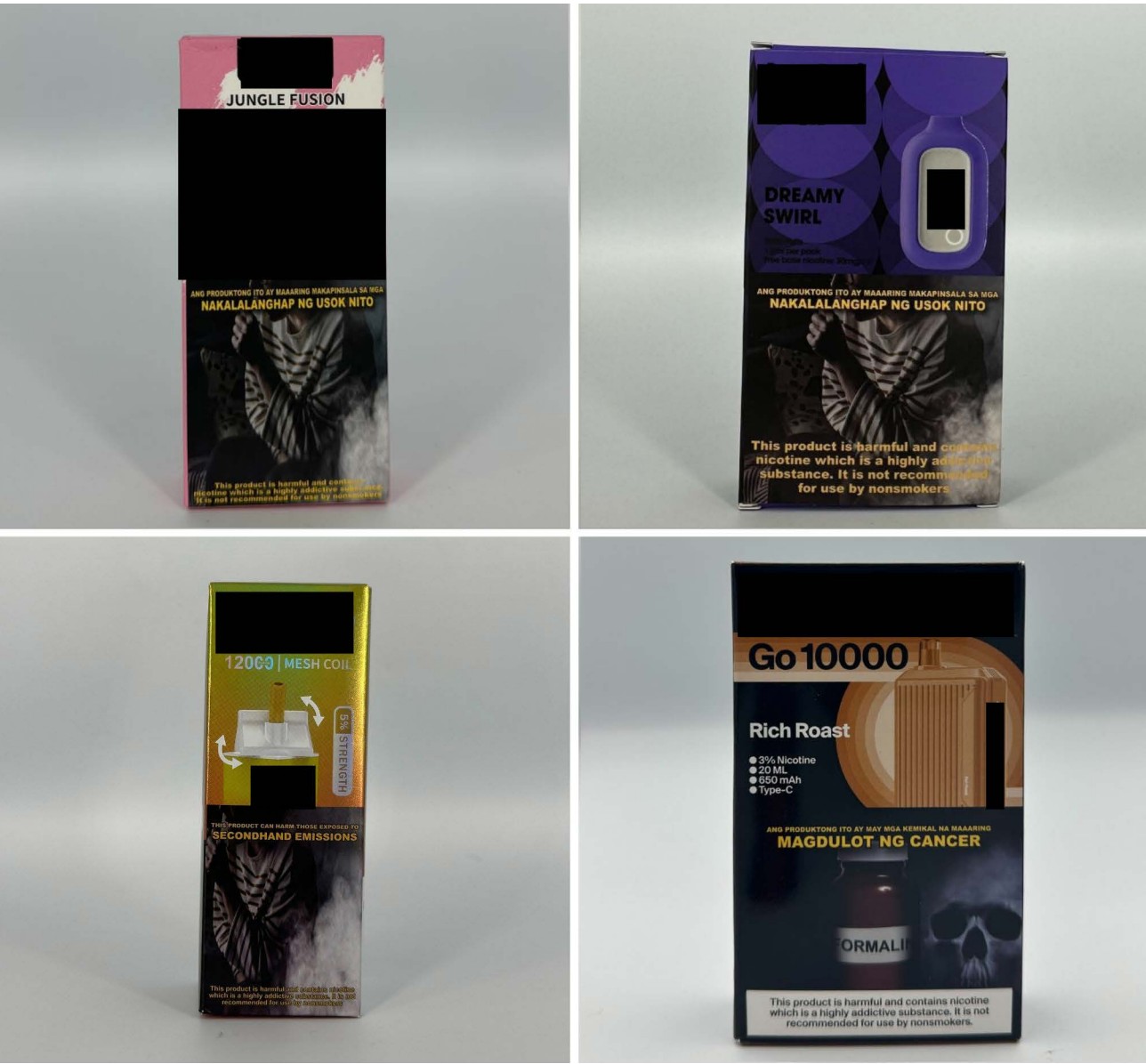

**Fig 4. Concept descriptors.**

the product packaging. In some cases, the verification scratch-off sticker redirects users to a website page about the product, rather than to an actual product verification page.

## Observations on implementation

Our team noted down several observations on the implementation of the Vape Law, including the presentation of flavor descriptors, age-verification mechanisms, and product testing.

## Flavor descriptor presentation

The Vape Law prohibits "unduly appealing" flavor descriptors that are proven to appeal to minors, particularly if they reference a "fruit, candy brand, dessert, or cartoon character"

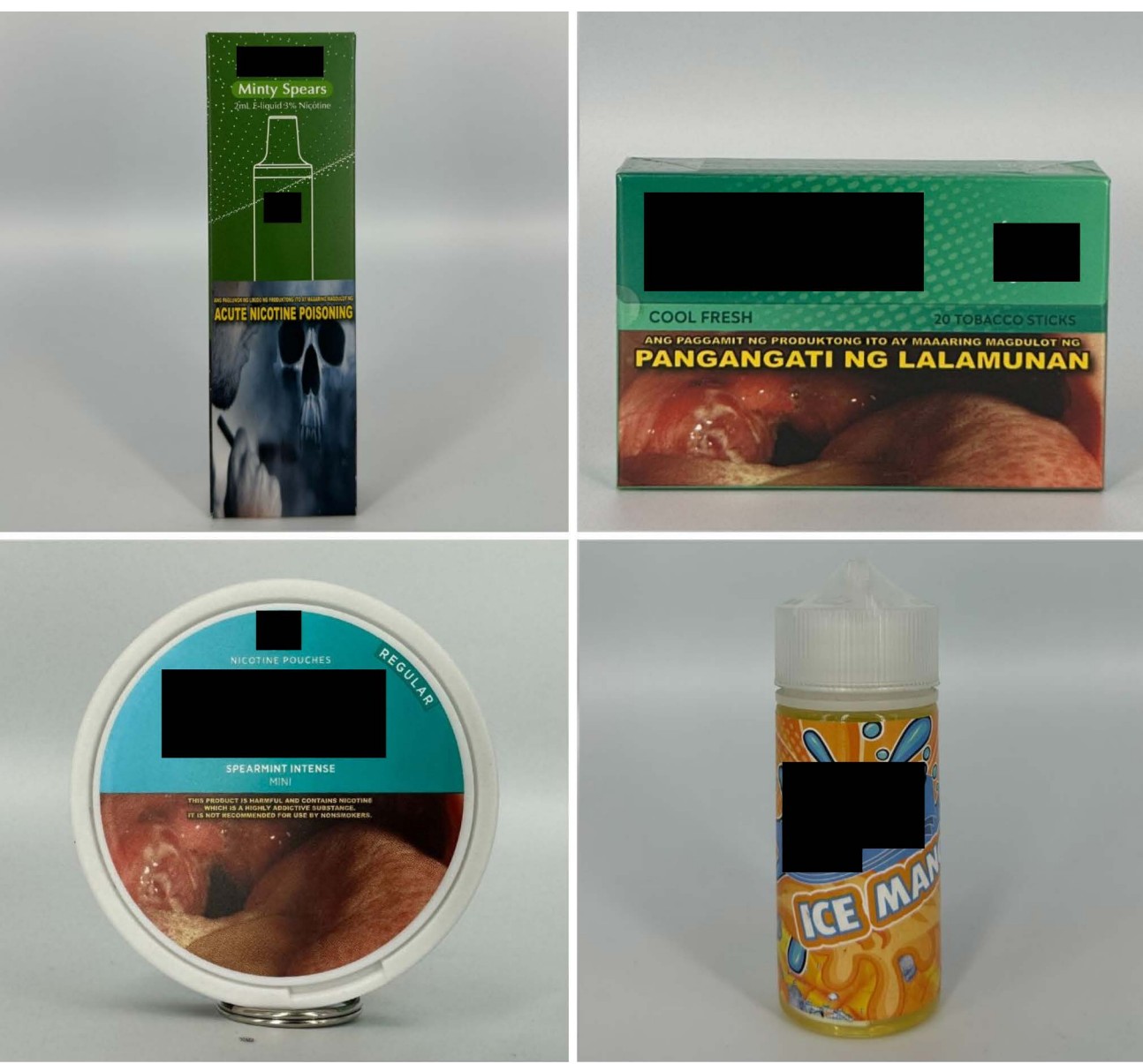

**Fig 5. Menthol flavor descriptors.**

[6]. We inconsistencies in the presentations of flavor descriptors, which may be a response to complying with the ban on unduly appealing flavor descriptors stipulated under the Vape Law. For example, some e-cigarette packs had original flavor descriptors covered by stickers with another flavor descriptor (e.g., using a sticker indicating "luscious green" to cover the original flavor descriptor "avocado milk") (S5 Fig). In other instances, the packaging of an e-cigarette device will feature an "unduly appealing" flavor descriptor, but upon checking the device, there is a different flavor descriptor indicated (S6 Fig). For example, external packaging for a disposable e-cigarette indicated "BBALL" as the flavor descriptor, but the flavor descriptor indicated on the device was "Butterball." Codebooks, booklets, or laminated menus of available e-cigarette flavors provided to the buyer by the sales representative, were often available to decode the true flavors for coded e-cigarette products.

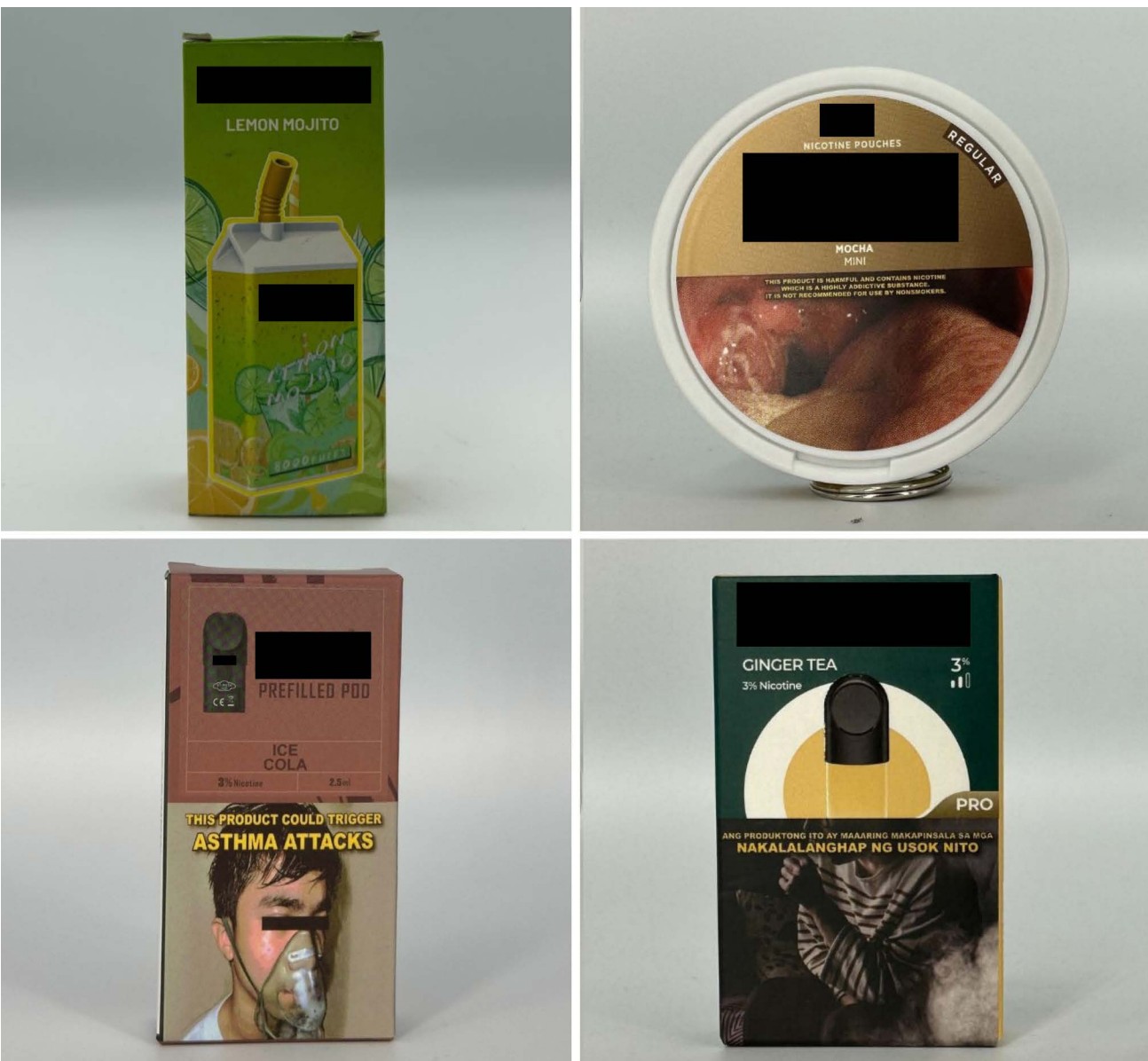

**Fig 6. Beverage flavor descriptors.**

### Absence of age-verification

The Vape Law mandates all retailers to verify the age of buyers by requiring the buyer to present any valid government-issued identification card. However, our team did not encounter any instance in which a retailer required us to present an identification card to verify our age.

### On-the-spot product testing

In several instances, we were required to test an e-cigarette product before purchase to ensure its functionality, even though this occurred indoors in areas where e-cigarette use is prohibited under the Vape Law.

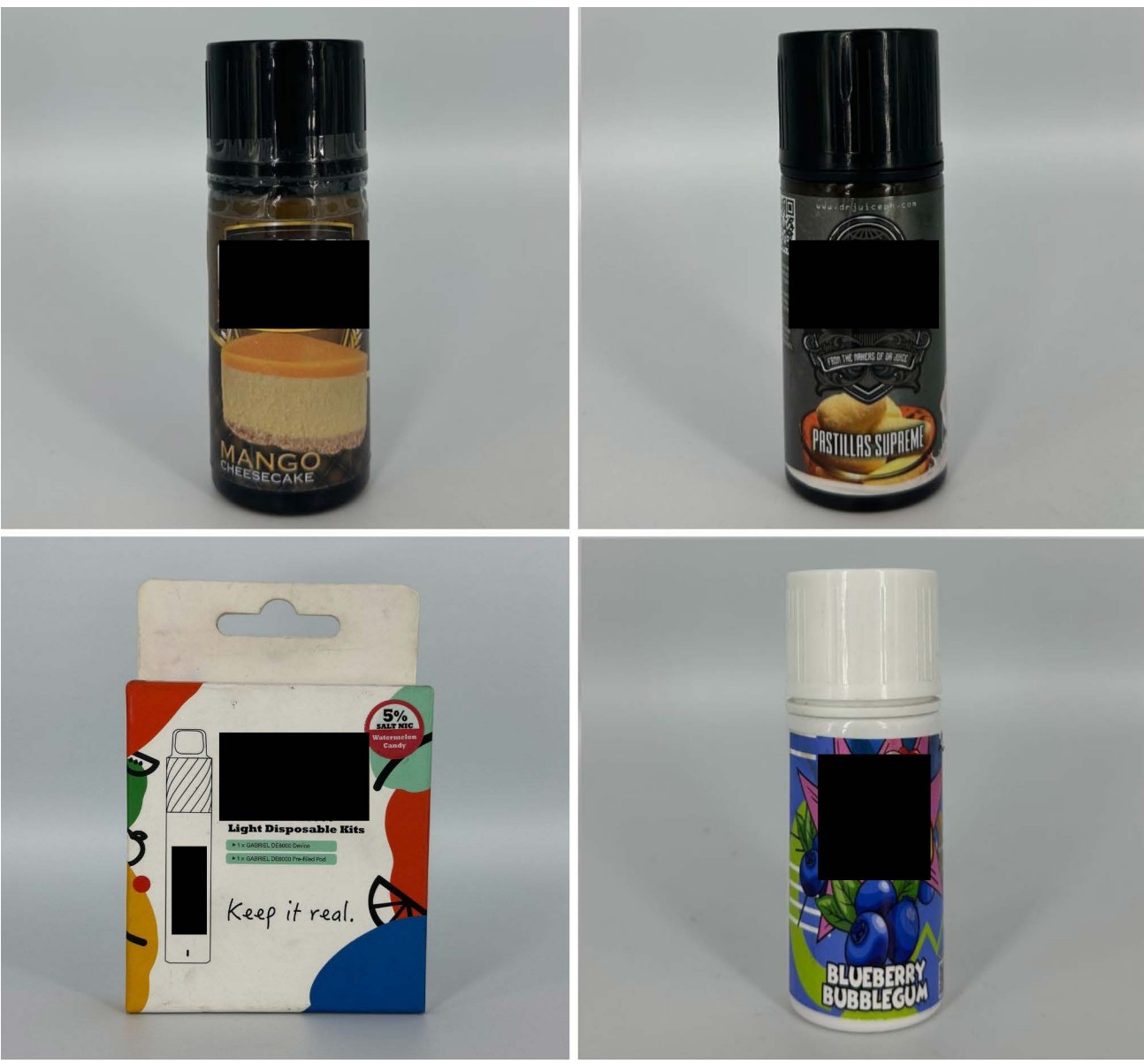

**Fig 7. Sweet flavor descriptors.**

## Discussion

We identified 11 categories of distinct flavor descriptors amongst a sample of 278 vapor products, vapor product systems, refills, heated tobacco products, and novel tobacco products. In order of frequency, these 11 categories are colors, fruits, codes or acronym-like descriptors (e.g., MCM for mocha cookie milkshake), concept descriptors, menthol, beverages, dessert/sweet/candy, miscellaneous, pop culture, tobacco, and nuts. Multiple flavor descriptors fit into more than one category, lending to 363 flavor descriptors across all categories. Our team observed implementation gaps in the Vape Law throughout the data collection process such as the availability of codebooks for coded e-cigarette products and the absence of age-verification mechanisms.

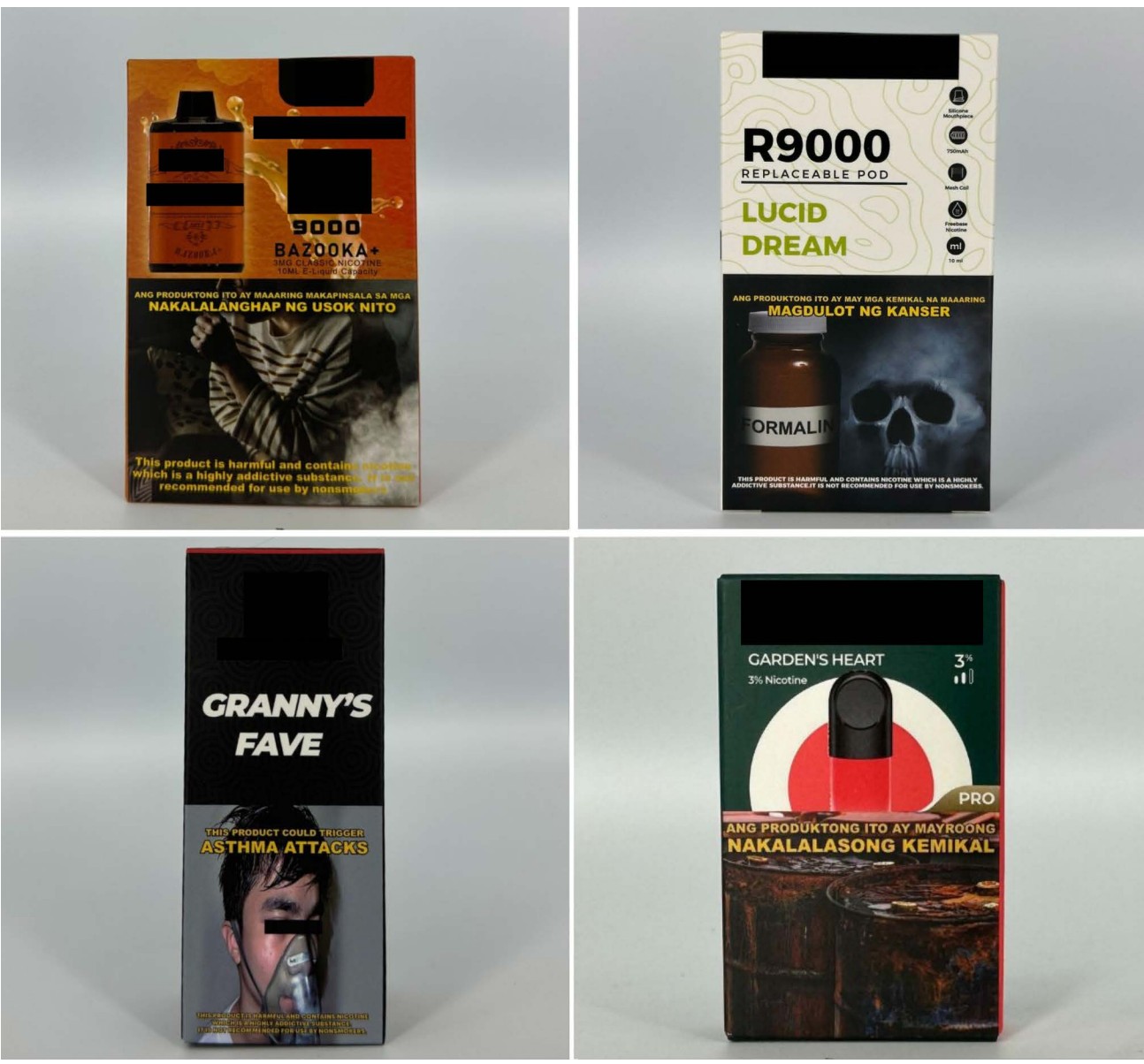

**Fig 8. Miscellaneous flavor descriptors.**

Given our findings, there are challenges and opportunities in implementing flavored e-cigarette, HTP, and novel tobacco product regulations. While there are challenges with product innovation and the policy loopholes of the Vape Law, there are opportunities to regulate and ban flavor descriptors, flavor imagery, and flavor substances.

## Challenges with product innovation

Our findings confirm that the e-cigarette, HTP, and novel tobacco product industry is continuously innovating flavors beyond traditional tobacco or menthol flavors. We discovered flavor categories such as *fruit*, *sweet*, and *menthol* that typically appeal to youth or young adults [34]. Young adult e-cigarette users prefer sweet flavors over non-sweet or flavorless

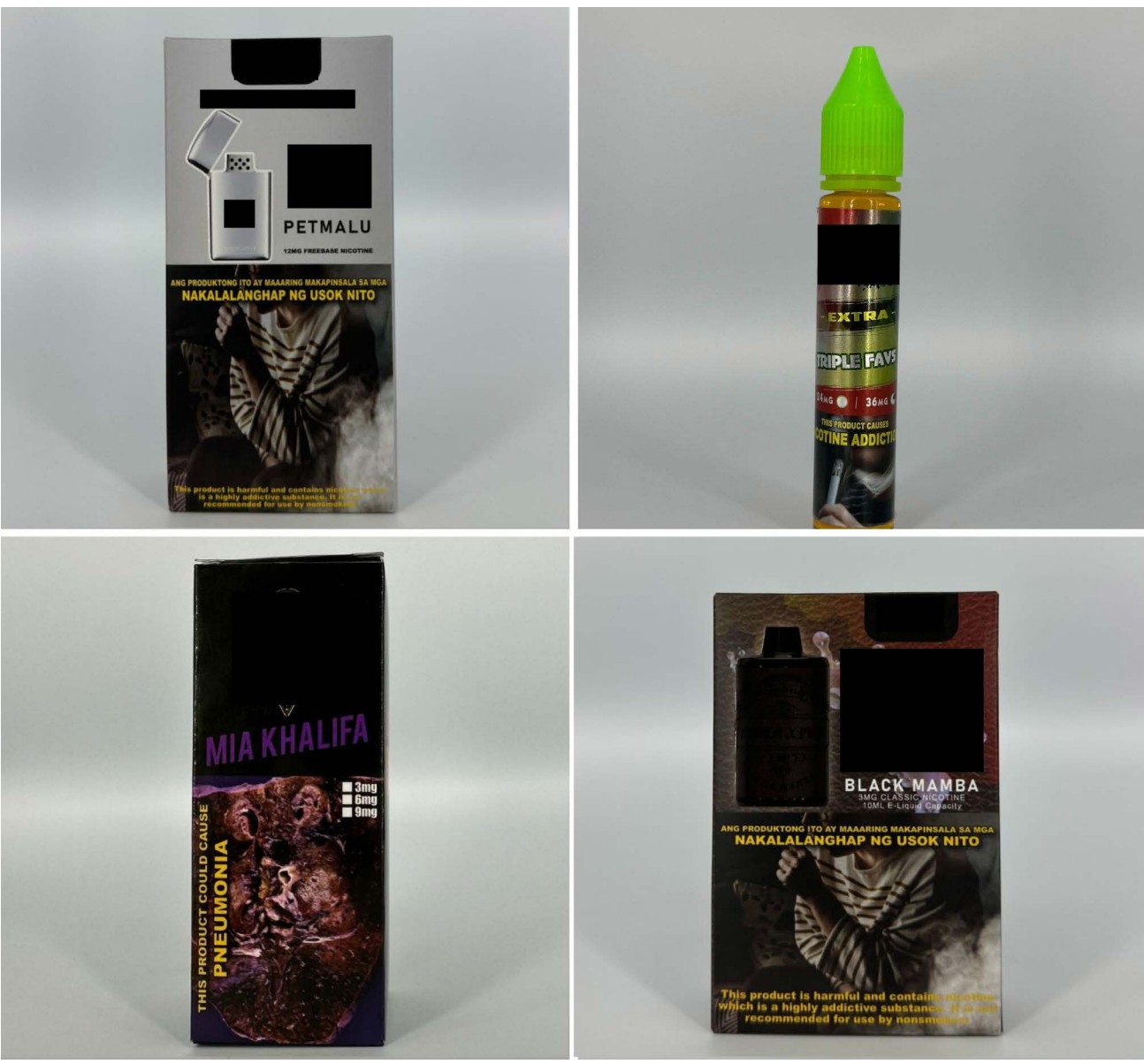

**Fig 9. Pop culture flavor descriptors.**

e-cigarettes [14]. Additionally, the youth are more likely to use non-tobacco e-cigarette flavors such as fruit and candy [35]. In our study, *fruit* and *candy* ranked second and seventh respectively while *tobacco* was ranked tenth. This high ubiquity of youth-appealing flavors warrants concern for public health—especially when contextualized in common youth and adolescent perceptions of e-cigarettes.

Adolescents perceive flavors such as menthol, candy, or fruit as less harmful than tobacco flavors [36], which can lead to more frequent e-cigarette use and nicotine dependence [37]. This is concerning considering that nicotine is particularly strong in children and young adults [38]. Flavored e-cigarettes are also appealing to current smokers looking to cut down on smoking [39]. A 2019 study found that young adult smokers enjoy fruit, candy, dessert, or menthol/mint flavors, which help them reduce smoking by increasing e-cigarette use [40].

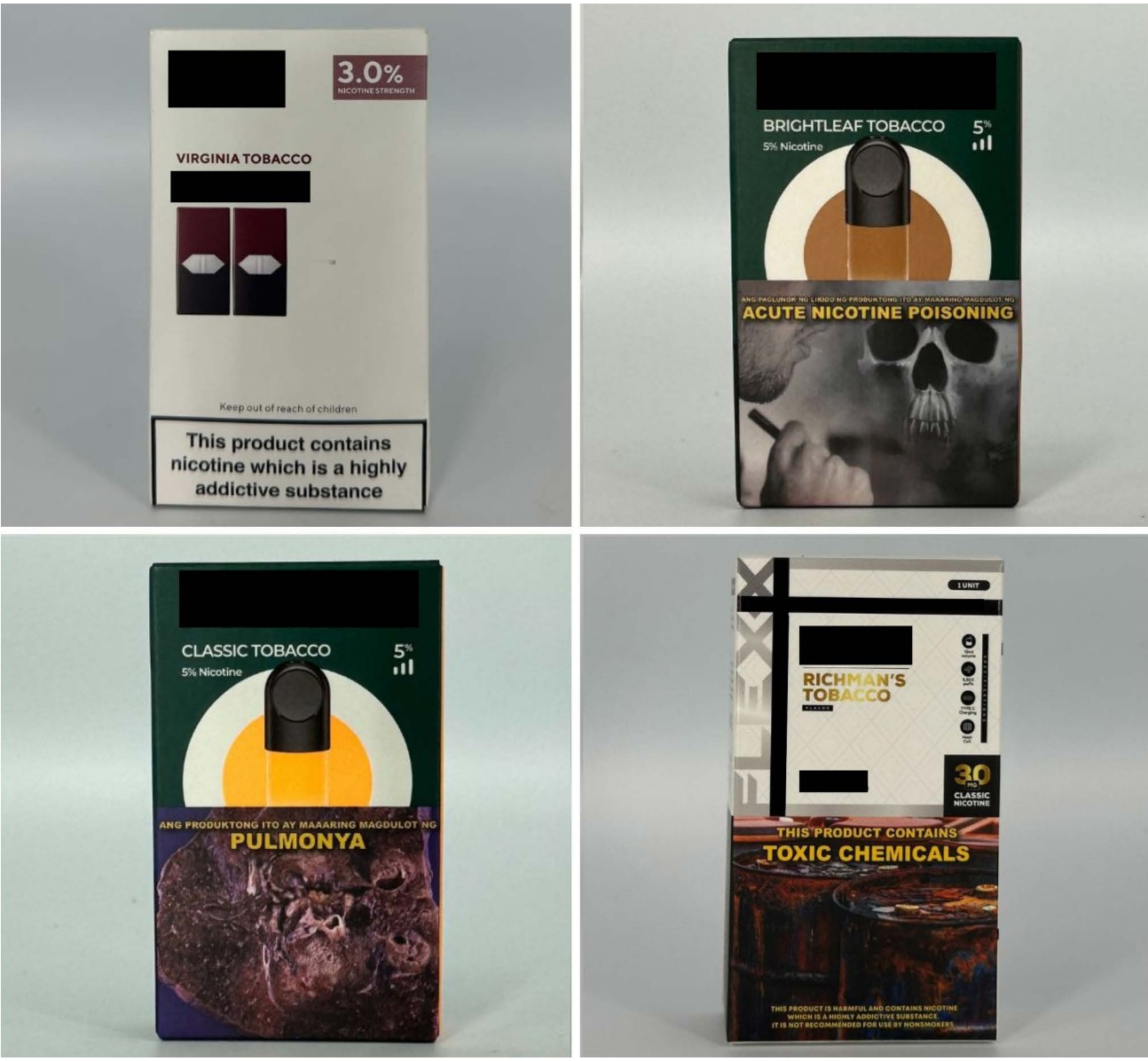

**Fig 10. Tobacco flavor descriptors.**

Flavors boost the rewarding value of e-cigarette use by making e-cigarettes more enjoyable while also boosting the reinforcing value by encouraging more effort to use them, increasing their addictive potential and abuse risk [41]. Additionally, nicotine is considered a gateway drug, often leading to using other substances such as illegal drugs like marijuana [42]. Using e-cigarettes with nicotine can lower the threshold for addiction to other dangerous substances, increasing susceptibility to substance dependency [43].

### Challenges with implementing the Vape Law: Compliance gaps and policy loopholes

Our team observed several compliance gaps and policy loopholes in implementing the Vape Law. One notable compliance gap is the proliferation of Vape Law-banned flavor descriptors.

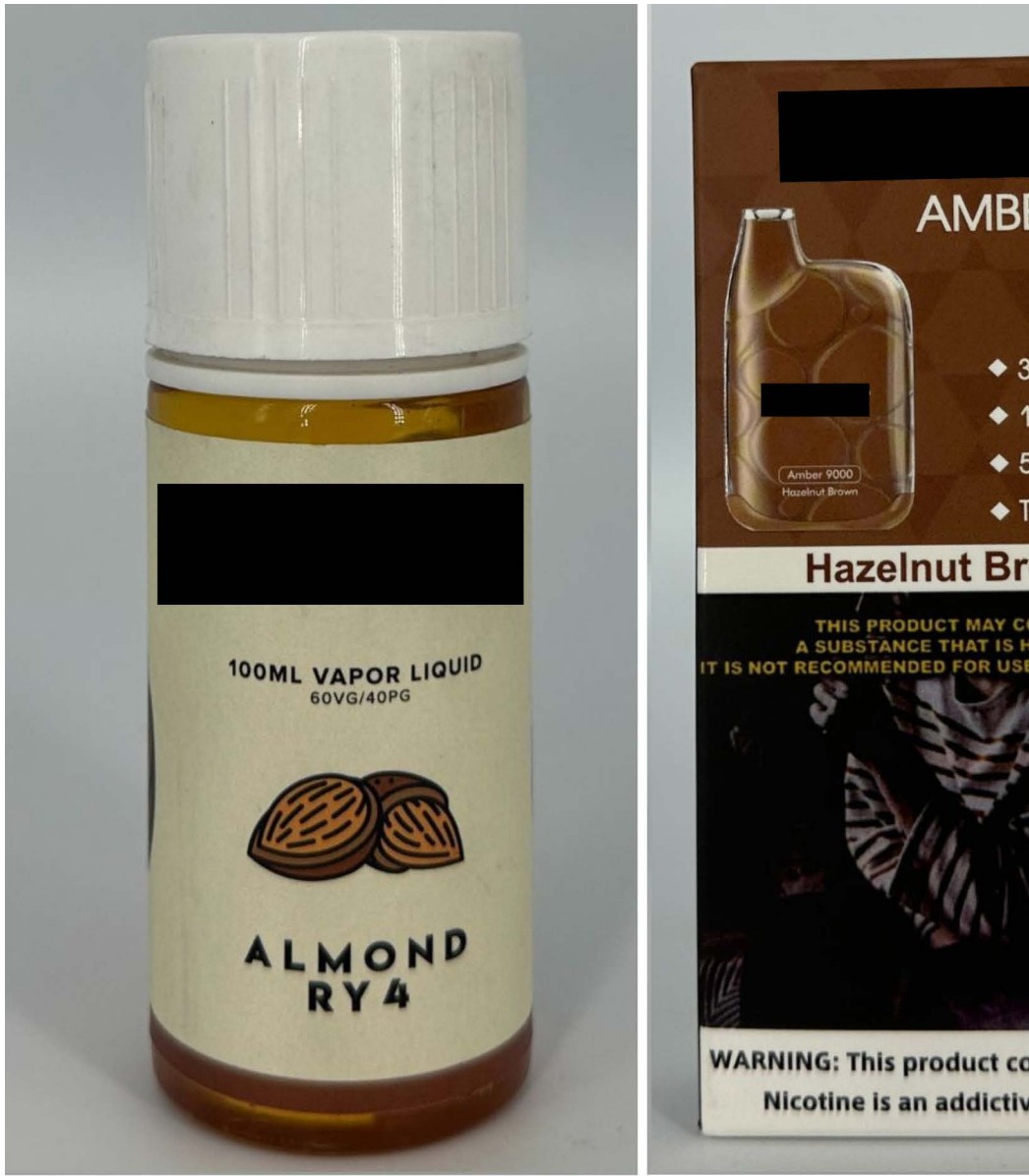

**Fig 11. Nut flavor descriptors.**

For example, under the Vape Law, "unduly appealing" flavor descriptors that "include a reference to a fruit, candy brand, dessert, or cartoon character" are prohibited. On the contrary, our team observed that this provision is yet to be fully complied with. We discovered flavor descriptors that specifically reference fruits, candy brands, desserts, and cartoon characters. *Slang words* also reference phrases that are appealing to minors yet are not prohibited under the Vape Law. Other categories can appeal to other demographics such as adults through *alcoholic beverage* or *coffee* flavor descriptors. Referring to popular *people* through *pop culture* flavor descriptors can evoke strong emotions within demographics—for example, by evoking the Filipino's enthusiasm for the sport of basketball by releasing an e-cigarette flavor called "Black Mamba."

On the other hand, our team observed several policy loopholes in the Vape Law. Some categories of flavor descriptors exploit loopholes in the Vape Law. For instance, products circumvent using "unduly appealing" flavor descriptors by using *colors* and *codes* as flavor descriptors. *Color* and *code* flavor descriptors are usually *sweet* or *fruit* flavors; yet, these flavors are concealed by the descriptor to comply with the Vape Law. Despite this, these two categories do not necessarily reduce the appeal of these products. *Colors* play a vital role in boosting the appeal of e-cigarette products [44,45], especially for minors who may have preferences for certain colors. Additionally, flavor descriptor *codes* can be easily deciphered by codebooks that consumers can peruse to discover the flavor of a coded, flavored product.

The ambiguity of the Vape Law's repealing clause also lends space for mixed interpretations of the law. The repealing clause is a catch-all statement that repeals all existing policies that govern vaporized nicotine and non-nicotine products and novel tobacco products. With no mention of which specific policies or regulations are repealed, the common interpretation could be that all previous policies—such as the flavor ban of FDA A.O. 2021-1069—are void. Despite this, businesses have not complied with the ban of "unduly appealing" flavor descriptors of the Vape Law. This is evidenced by the ubiquity of flavor categories such as fruits, sweets, or pop culture that we identified.

Another notable loophole is the lack of regulation on the shapes of e-cigarette devices and systems. E-cigarette devices shaped as energy drinks, juice boxes, or milk cartons are very common. This can directly increase the appeal of e-cigarettes to children and the youth. With no regulation on device shapes in the Vape Law, this presents an opportunity for the e-cigarette industry to actively target younger demographics.

Lastly, freebies and promotional tactics such as branded lanyards or social media handle plugs directly violate the prohibition of promotional merchandise under the Vape Law. Promotional merchandise can be viewed as a form of "brand stretching" or "trademark diversification" where brand names are used on non-tobacco or non-e-cigarette products [33]. These promotional items continue to circulate particularly in e-cigarette and HTP businesses and can boost the novelty and appeal of flavored e-cigarettes and HTPs.

While the Vape Law sets regulations for vaporized nicotine and non-nicotine products, its challenges lie in its implementation. There is a need for comprehensive education amongst retailers on the stipulations of the Vape Law, along with its corresponding consequences for non-compliance. However, this cannot be achieved without enforcement bodies. One way to consider boosting enforcement is by engaging local government units and health agencies in monitoring Vape Law implementation and compliance. These bodies must conduct regular inspections and penalize violators accordingly. Efforts to achieve this are already being carried out by regional bodies such as the Metropolitan Manila Development Authority, but gaps remain in workforce capacity. Local governments should also consider introducing local task forces and volunteer brigades within communities and around schools to ensure compliance at the *barangay* level (the most basic governance level in the Philippines).

## Opportunities to regulate flavor imagery

Our team observed that flavor imagery, closely linked with flavor descriptors, is common, particularly in e-cigarettes. These often include attractive images that directly reference specific flavors. However, flavor imagery is not regulated by the Vape Law. This lack of regulation has several implications, including the potential of flavor imagery to attract consumers, undermine health warnings on packaging, and influence harm perceptions. Existing evidence shows that sweet and fruit e-cigarette imagery interferes with warning labels and increases cue-reactivity and appeal as compared to tobacco flavors [46,47]. Cue-reactivity includes the physiological, psychological, or cognitive responses related to reward anticipation [48].

## Opportunities for a flavor ban and regulations on accessibility

The WHO recommends several policy options to regulate e-cigarettes catered to existing interventions implemented by countries [25,26]. For countries such as the Philippines that permit the sale, importation, distribution, and manufacture of e-cigarettes, recommended interventions include a flavor ban, taxation, and limiting the concentration and quality of nicotine.

Incorporating the recommendations of the WHO entails amending the Vape Law to clarify the nature and extent of prohibited flavored e-cigarettes, HTPs, and novel tobacco products. This can include further defining which flavor descriptors are banned or prohibited. For example, flavor descriptors, such as *colors* and *concept descriptors*, that are not deemed "unduly appealing" under Vape Law but still promote the appeal of e-cigarettes must be banned. The use of colors for packaging designs and as flavor descriptors must be closely regulated to reduce any consumer biases. *Code* flavor descriptors and the availability of code-books should be accounted for in these amendments.

When implementing a flavor ban, the flavor ban must be a complete ban of flavorings along with flavor descriptors and flavor imagery. Multiple studies have confirmed that e-cigarette users are more likely to quit or significantly reduce using e-cigarettes under a flavor ban [40,49,50]. While flavor bans can reduce e-cigarette use, poor regulations on the accessibility to e-cigarette products along with retailer non-compliance can hamper these efforts [49,50].

Strict enforcement is critical to implementing successful effective e-cigarette regulations. Through our data collection process, we observed several gaps and loopholes in the Vape Law implementation. For example, most retailers provide a codebook or flavor portfolio that allows the buyer to see what flavors certain codes pertain to. We did not encounter any retailer who asked for forms of identification before purchase. Additionally, flavor imagery continues to remain common despite its instrumental role in flavored e-cigarette, HTP, and nicotine pouch appeal. With these existing gaps, it is possible for minors not only to gravitate toward flavored products but to easily access them as well. One way to prevent this is by strengthening age-verification interventions. Retailers believe there is little to no risk of noncompliance due to irregular law enforcement compliance checks [51]. Enforcing age verification mechanisms necessitates a shift in retailer behavior. Targeted training sessions on the legal repercussions of selling to minors and the health risks of e-cigarette, HTP, and nicotine pouch use could help achieve this. Since many retailers are users themselves, these initiatives may foster a deeper commitment to compliance and the effective implementation of age-verification practices.

Taxation is the most effective measure to reduce e-cigarette, HTP, and nicotine pouch use. While taxation may not directly relate to the findings of our study, a comprehensive policy strategy for e-cigarette, HTP, and nicotine pouch prevention and control should encompass not only the regulation of flavors and branding but also the overall availability, accessibility, and affordability of these products. With this in mind, taxes should be integral to any policy addressing both flavored and non-flavored products. Increasing the excise tax on e-cigarettes can reduce their affordability, especially for the youth [52]. At present, the Philippines imposes a 12% VAT on the sale and importation of e-cigarettes [53]. However, the WHO recognizes excise taxes are ideal as they "raise the relative price of tobacco and nicotine products compared to other products and services" [52]. An excise tax imposition, specifically on an ad valorem basis, can further limit consumption, discourage initiation, and prevention of related diseases [53].

Another way forward is to introduce plain packaging, forcing the industry to remove logos, colors, brand images, or promotional information on packaging other than brand names and product names displayed in a standard color and font style [54]. Plain packaging can be an

effective method to eliminate packaging as a marketing tool, to reduce the attractiveness and appeal of flavored products, and to increase the noticeability and effectiveness of health warnings. In turn, this eliminates the deceptive appeals that come with products created through flavor descriptors and imagery.

Lastly, there is a pressing need for seamless integration and clear communication of e-cigarette, HTP, and novel tobacco product flavor regulations. The FDA AO 2021-1069 initially banned all e-cigarette flavorings and additives except tobacco and menthol, but the Vape Law has introduced regulatory ambiguity. Specifically, the Vape Law's repealing clause, intended to overturn previous regulations on e-cigarettes, HTPs, and novel tobacco products, fails to clarify the status of the FDA's flavor ban, leading to differing interpretations about its enforceability. As a result, businesses might argue that the FDA flavor ban contradicts the Vape Law, which focuses on regulating specific flavor descriptors rather than flavorings. This confusion is further compounded by the mandated transfer of regulatory jurisdiction for vaporized nicotine and non-nicotine products and novel tobacco products from the DOH FDA to the DTI. The regulation of these products must be managed by the corresponding authority on health concerns, especially as cases of EVALI continue to rise around the world and in the Philippines [55]. Hence, it is prudent to reauthorize the DOH FDA to regulate e-cigarettes, HTPs, and novel tobacco products beyond only consultation matters on consumer safety. Strengthening enforcement through collaboration with local government units and health agencies is crucial, which should involve regular inspections and penalties for violations.

## Limitations of the study

One limitation of our study was our reliance on retailer-provided codebooks to identify certain product flavors. In some cases, there were inconsistencies between the packaging and the device, with flavor descriptors missing from the device. Additionally, when codebooks were unavailable, we could not always identify the flavors, potentially affecting the completeness of our results.

Additionally, due to budgetary constraints, our team was unable to purchase all unique products from each retailer we visited, particularly for e-cigarettes and refills. This constraint is also amplified by the high diversity of product variations offered by retailers. However, we ensured the representation of our sample by following the TPackSS protocol as closely as possible; for instance, purchasing at least one sample per every available brand of product at the vendor. Additionally, this study did not delve into consumer perceptions and preferences of e-cigarette, HTP, and nicotine pouch flavors in the Philippines. Future studies can explore consumer perceptions and consumer behavior to inform existing control efforts and regulations.

## Conclusion

The growing diversity of local e-cigarette, HTP, and nicotine pouch flavors demands for stricter regulation to safeguard public health, especially among children and the youth. Restoring the nontobacco and nonmenthol flavor ban stipulated in FDA A.O. 2022-1069 can reduce the variety and diversity of available flavored e-cigarette, HTPs, and nicotine pouch products. This ban should include comprehensive regulations or complete bans on flavorings, flavor descriptors, and flavor imagery. However, a flavor ban must be accompanied by rigorous retailer compliance policies free from policy loopholes. Increasing taxation and implementing plain packaging can limit the accessibility of these products and eliminate appeals that drive consumer demand.

## Supporting information

**S1 Table. Neighborhood data collection sites.**
(XLSX)

**S2 Table. Definition of terms.**
(DOCX)

**S3 Table. Type of stores purchased from.**
(XLSX)

**S4 Table. Manufacturer distribution of samples.**
(XLSX)

**S1 Data. Data capture form, codebook, data sheet, and categorized flavor profile.**
(XLSX)

**S1 Fig. Samples of flavor imagery.**
(TIFF)

**S2 Fig. Iconography indicating the flavor intensity and nicotine level.**
(TIFF)

**S3 Fig. E-cigarette product device shapes.**
(TIFF)

**S4 Fig. Merchandise acquired from production promotions and inclusions.**
(PNG)

**S5 Fig. Flavor descriptor cover-ups.**
(PNG)

## Acknowledgments

We would like to acknowledge Eunice U. Mallari and John Rafael Arda for their technical assistance in crafting the research protocol for this study.

## Author contributions

**Conceptualization:** Samantha Joan Ackary, Patrik James De leon Cabrera, Alen Josef Santiago, Gianna Gayle Amul.

**Data curation:** Samantha Joan Ackary, Patrik James De leon Cabrera, Alen Josef Santiago.

**Formal analysis:** Samantha Joan Ackary, Patrik James De leon Cabrera, Alen Josef Santiago.

**Methodology:** Samantha Joan Ackary, Patrik James De leon Cabrera, Alen Josef Santiago.

**Project administration:** Samantha Joan Ackary, Patrik James De leon Cabrera, Alen Josef Santiago.

**Software:** Samantha Joan Ackary, Patrik James De leon Cabrera, Alen Josef Santiago.

**Supervision:** Samantha Joan Ackary, Patrik James De leon Cabrera, Alen Josef Santiago.

**Validation:** Samantha Joan Ackary, Alen Josef Santiago, Gianna Gayle Amul.

**Writing – original draft:** Samantha Joan Ackary, Patrik James De leon Cabrera, Alen Josef Santiago.

**Writing – review & editing:** Samantha Joan Ackary, Patrik James De leon Cabrera, Alen Josef Santiago, Gianna Gayle Amul.

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
