## [Decision Letter · Decision Letter 0]

23 Sep 2024

PGPH-D-24-01748

Examining the Flavors of E-Cigarettes, Heated Tobacco Products, and Novel Tobacco Products in the Philippines: Regulatory Challenges and Opportunities

Dear Dr. Ackary,

Thank you for submitting your manuscript to PLOS Global Public Health. After careful consideration, we feel that it has merit but does not fully meet PLOS Global Public Health’s publication criteria as it currently stands. Therefore, we invite you to submit a revised version of the manuscript that addresses the points raised during the review process.

We look forward to receiving your revised manuscript.

Kind regards,

Chandrashekhar T. Sreeramareddy

Academic Editor

Journal Requirements:

1. In your Methods section, please provide additional information regarding the permits you obtained for the work. Please ensure you have included the full name of the authority that approved the field site access and, if no permits were required, a brief statement explaining why.

Additional Editor Comments (if provided):

Dear Authors,

The manuscript timely and relevant in the context of growing and rapidly evolving e-cigarette marker. The reviewers were favorable on the manuscript. However, the the comments made by all three reviewers are to improve the content of the manuscript.

look forward to receive the revised manuscript in near future.

Reviewers' comments:

Reviewer's Responses to Questions

**Comments to the Author**

1. Does this manuscript meet PLOS Global Public Health’s publication criteria ? Is the manuscript technically sound, and do the data support the conclusions? The manuscript must describe methodologically and ethically rigorous research with conclusions that are appropriately drawn based on the data presented.

Reviewer #1: Yes

Reviewer #2: Yes

Reviewer #3: Yes

2. Has the statistical analysis been performed appropriately and rigorously?

Reviewer #1: Yes

Reviewer #2: N/A

Reviewer #3: N/A

3. Have the authors made all data underlying the findings in their manuscript fully available (please refer to the Data Availability Statement at the start of the manuscript PDF file)?

Reviewer #1: No

Reviewer #2: Yes

Reviewer #3: Yes

4. Is the manuscript presented in an intelligible fashion and written in standard English?

Reviewer #1: Yes

Reviewer #2: Yes

Reviewer #3: Yes

5. Review Comments to the Author

Reviewer #1: While the data analysis is mainly descriptive, the study collects relevant primary data on an important public health topic, which is appropriately analyzed considering local regulations and best practices in the world. Below I provide some minor suggestions that I hope will be useful.

• Title, line 3. Please consider specifying flavor descriptors/representation instead of just flavor in the title; may be more precise.

• Introduction, lines 37-38. Please indicate whether this increase is statistically significant (confidence intervals or a p-value would be informative).

• Methods,

o Lines 91-92 only mention cigarette and cigar packs but not ENDS, which are the focus of the analysis. Therefore, it is unclear whether only ENDS or both cigarettes and ENDS were collected. The information provided in these lines should be consistent with that in the sample section.

o Lines 94-98. Please clarify what data were unavailable, just socioeconomic status? On lines 95-96 please be specific about the data used and the sources.

• Results

o Lines 140, 149. Information about the sample size is inconsistent: line 122 indicates 313, line 140 indicates 280 and line 149 indicates 315. On line 140 it should be specified that 280 corresponds to products with flavor descriptors.

o Line 142: although this is explained later in the draft, the category “codes” should include an example in parenthesis such as (e.g. acronyms).

o Line 151. Table 1 seems unnecessary as all the information it provides is summarized on line 149 (note that the % differs, it should be the same). I suggest merging tables 1 and 2.

o Line 156. Why are mall kiosks the main place of purchase? Is it because this is the most common point of sale in the country? Is this result (a dominant point of sale) common when using the TPackSS walking protocol?

o Line 162. Table 4 is very extensive and although it presents relevant information, it could be summarized by grouping by country of origin. It would also be interesting to understand how this information relates to the central point, flavor descriptors, specifically, whether there are differences (or not) in the use of flavor descriptors between domestic and foreign producers.

o Lines 219-220. This part about marketing tactics is interesting (specially the discussion in lines 301-306) but the methods do not explain that this type of information was also collected; should be specified.

• Fig 1 to 11, if of good quality, are an interesting addition to help understand the variety of flavor descriptors. But 4 photos from each of the 11 categories may be too much. Maybe it is an editorial decision.

Reviewer #2: 1.　Lines 40-41, ” About 25% of Filipino students aged 13–15 have tried e-cigarettes while 14% currently use them”

At what age can you use e-cigarettes in the Philippines? Also, is there any age confirmation at the retailers?

2. Lines 57-60, It mentioned the Vape law, but I could not understand why the Vape law was to be enforced. If you know any reason, it would be easier to understand the story if you had. For example, political or economic reasons.

3. Do all e-cigarettes sold in the Philippines contain nicotine? I know there are products that do not contain nicotine, but is the legal requirement different depending on whether they contain nicotine or not? For example, age restrictions, sales locations, etc.

4. Following up on the above question, I thought it would be informative if there were products with and without nicotine in e-cigarettes, what the proportions are with and without nicotine, and whether there is a difference in flavor, etc.

5. Lines 347-349, You mentioned age-verification, but what exactly do you think you need to do to get retailers to follow your method?

6. Even though there are loopholes in the Vape law, there is still some degree of regulation, and your investigation showed that this is still not being followed. What do you think is the first step to make sure that the standards of the Vape law are followed?

Reviewer #3: Thank you for the opportunity to review your manuscript. This study provides valuable insights by adapting the TPackSS protocol to characterize flavor descriptors on e-cigarette packaging in the Philippines—a critical topic given the recent policy changes surrounding e-cigarette flavors in the country. However, the manuscript would benefit from improved structure and enhanced clarity in its methodology, as well as greater precision and consistency in the use of terminology. In particular, clearer definitions for product types and flavor descriptor categories, as well as more detailed descriptions of the methods used to examine flavor imagery, marketing strategies, and implementation observations, are needed. Below, I’ve provided specific suggestions to help improve the manuscript for your consideration.

ABSTRACT

Line 20-21: Instead of the positive adjectives, consider more precise descriptions and focus on how these products are perceived. For example, “perceived novelty” instead of “esteemed novelty”, “ever-changing flavors” instead of “innovative flavors”, “sensations that are attractive to consumers” instead of “pleasing sensations.” The Institute for Global Tobacco Control’s video on language considerations might help guide word choice: https://youtu.be/afd-ciayb5U

Line 27: It’s unclear what “codes” mean here. Some explanation would be helpful.

There seems to be a disconnect between the results (lines 26-29) and conclusions (lines 30-34). While the study aims, methods, and results presented are around descriptors, the authors also make recommendations for imagery regulation and taxation policy. It might be helpful to add a connecting sentence between the results and conclusions for a smoother transition, and ensure all recommendations are linked to the study results presented.

INTRODUCTION

Line 36-44: Focus on e-cigarettes instead of other tobacco products. Consider removing the sentence on tobacco-related deaths (line 36) and add more on harms associated with e-cigarettes.

Line 45-50: This paragraph should be expanded to include a more comprehensive discussion of the literature on e-cigarette flavor appeal, particularly among youth. It would also be beneficial to reference relevant studies conducted in the region.

Line 46: The abbreviation “ENDS” only appears once in the whole article. Might not be needed.

Lines 51-56: Rather than discussing general e-cigarette regulations, this paragraph could focus on an overview of e-cigarette flavor bans across countries, and elaborate on the associated challenges and considerations by providing more details.

Line 59: Add the year for Republic Act 11900.

Line 61: Flavor substances are introduced here for the first time. It would be helpful to establish this concept earlier by discussing regulations on both flavor substances and descriptors in the previous paragraph when overviewing e-cigarette flavor bans. Additionally, please clarify whether the 2021 ban regulated flavor substances.

Line 63-65: This sentence is confusing and unclear. The phrase "implementation ambiguities" is not well explained. Additionally, the connection between these ambiguities and the Vape Law's repealing clause is not clearly articulated. It would be helpful if the authors clarified what they mean by "implementation ambiguities", provided a more detailed explanation of how the repealing clause contributes to these ambiguities, and included a relevant citation to support this point.

Line 77-79: It is unclear what “mixed implementation” means. In addition, some elaboration on “significant opportunity to strengthen flavored and unflavored e-cigarette regulation” would be helpful.

Line 80-82: The study cited here does categorize e-cigarette flavors. It might be more accurate to state “flavored e-cigarettes available at points of sale.”

Lines 82-83: The stated study aim is quite vague and lacks precision. It would be helpful to provide more specific details on how e-cigarette flavors will be classified (e.g., descriptors on e-cigarette packaging at points of sale in the Philippines) and which elements of the policy landscape will be assessed.

METHODS

Line 86: The abstract states that the TPackSS protocol was adapted, while here it says adopted. Please clarify whether the TPackSS protocol was adapted or simply adopted. If adapted, explain the modifications made. There is no need to mention TPackSS protocol repeatedly in places where it is simply followed.

Line 92-94: Four cities in each of the four regions would be 16 cities in total. Please list all 16 cities, or correct the sentence to say “from the biggest city in each region.”

Line 94-96: “We chose 12 neighborhoods (barangays) for each city based on specific characteristics like population density, economic importance, and geographical location.” Please list all the neighborhood characteristics considered (using "including" instead of "like"), specify how these characteristics informed the sampling process (e.g., was the goal to achieve balanced distribution across each parameter?), and clarify how “economic importance” was determined. In addition, it would be helpful to include the list of neighborhoods selected in the supplemental material.

Lines 102-109: “Starting from a commercial hub in each neighborhood, our team walked for five minutes until we encountered one of the six specified vendor types. We then purchased one of each unique pack available for sale. […] In subsequent neighborhoods, our team purchased one of each unique pack not already obtained from previous vendors.” Please clarify “unique pack” of what. Additionally, clarify how the decision was made to proceed to the next neighborhood—at what point was data collection considered complete for one neighborhood? From the description provided, it appears that data collectors only visited one store per neighborhood, which would likely not yield a representative sample across vendor types. If this is not accurate, please correct and provide additional information.

Lines 116-118: "The codebook covers various packaging elements, including structural features (size and shape) and graphic elements (brand or flavor descriptors)." Please explicitly state the specific packaging elements examined in this study and explain how descriptors were defined.

Lines 118-120: "Two trained researchers conducted coding in two rounds to ensure result accuracy. A third researcher joined the second round of coding to address any discrepancies identified during the initial coding sessions." Please clarify what is meant by "two rounds" of coding. How many packages were coded during each round? What do "initial coding sessions" refer to, and what steps were taken after those sessions to complete the coding of all data?

Line 122: The term “e-cigarette” is used throughout other sections of the manuscript, but the term “vapor” is introduced here. For consistency, consider using the same terminology throughout.

Lines 122-129: Please define each product type clearly and use precise terminology. For instance, if “vapor products” only include pods and cartridges, consider using “e-cigarette pods/cartridges.” If “novel tobacco products” refer solely to nicotine pouches, use “nicotine pouches” instead. Additionally, use “heated tobacco product inserts” instead of “heated tobacco products” and “e-cigarette refills” instead of “refills” for greater precision.

Line 129: Nicotine pouch is mentioned here for the first time. If examining nicotine pouches is part of the study aim, please state this in the introduction section and provide the rationale, including relevant background information on nicotine pouches.

Line 126: The abbreviation “HTP” is introduced here for the first time. Please place it in parentheses following “heated tobacco product” when it is first mentioned earlier in the article. However, since lines 126 and 127 are the only places in the article where this abbreviation is used, it may not be necessary to retain it. If the authors choose to keep it, please replace “heated tobacco product” with “HTP” throughout the rest of the article for consistency.

Line 130-132: The inclusion criteria should be introduced earlier, before getting into the detailed steps of data collection.

Line 134-137: The analysis section requires significantly more detail. Please elaborate on the processes involved in the inductive and deductive approaches. Were inductive codes identified, and if so, how? Please also clarify whether the deductive approach refers to codes from the TPackSS codebook. In addition, it would be helpful to explain how flavor descriptors were defined and provide more information on the categorization process. Do the distinct codes and sub-codes refer to the codes generated through the inductive and deductive processes?

RESULTS

Line 139-147: This paragraph should be moved to the beginning of the discussion section.

Line 149: Consider revising this sentence to provide a more complete statement that includes the number of stores and the locations for data collection. For example, “Our team collected 315 products from XXX stores across a total of XX neighborhoods (barangays) in Quezon City, Antipolo City, Cebu City, and Davao City. Among these, 280 (88.89%) featured flavor descriptors.”

Tables 1 & 2: These tables could be combined by listing product types in the rows and providing breakdowns by flavor descriptors in the columns. Each cell can then include numbers for counts, followed by percentages in parentheses.

Line 156: Consider listing top 3 store types where products where purchased in the text, and remove Table 3 (or move to the appendix).

Table 4: Consider moving this table to the appendix.

Tables 5 & 6: Consider combining these two tables. Additionally, please use consistent terms for flavor descriptors in the tables and in the text. For example, “Fruits” is used in Table 5 and the text, whereas “Fruity” is sued in Table 6.

Line 172: The sentence "Color descriptors are denoted as standalone colors, adjectives, or nouns" is unclear. Please clarify or rephrase for better understanding.

Line 176-177: Was the study team always able to discern what the acronym-like mean? Some descriptions on how the determinations were made would be helpful.

Figures 1-11: Please provide figures with better image quality.

Line 193: Please specify whether steps were taken to confirm flavor descriptors such as RY4.

Line 194-196: How was it determined whether a descriptor is associated with a flavor, sensation, taste, or experience? Many examples in the miscellaneous category look like they could be concept descriptors.

Line 209-215: The results on flavor imagery should be presented more systematically, including image examples, rather than merely listing a few examples. Additionally, please provide details on how flavor imagery was examined earlier in the methods section.

Line 216-220: The results on marketing strategies should be presented more systematically, including image examples. Please specify that a wide range of flavors and device shapes resembling other objects were used, instead of referring to this as “innovative marketing.” Additionally, provide details on how marketing strategies were examined earlier in the methods section.

Line 221-321: The results on observations of implementation should be presented more systematically, including image examples. Additionally, please provide details earlier in the methods section on how implementation was examined, including information on the specific policies being assessed.

DISCUSSION

Line 238: Consider replacing “innovating flavors” with “introducing new flavors.”

Line 239-240: Include citations for the statement that fruit, sweet, and menthol flavors typically appeal to youth and young adults.

Line 247: Avoid starting a paragraph with "For example."

Line 255: Elaborate on the “rewarding and reinforcing value” of flavors.

Line 258: Elaborate on “getaway drug.”

Line 259: Clarify what “threshold for addiction” means.

Lines 264-268: The examples provided here indicate a failure in compliance with the law rather than actual loopholes, while some of the examples discussed later point to genuine loopholes in the Vape Law. It would be beneficial to clarify this distinction in the text. Additionally, consider discussing recommendations for addressing these compliance failures separately from the required policy changes to address the loopholes.

Line 339-342: The findings around retailer behaviors are mentioned here for the first time. These should be first introduced in the results section.

Line 350-357: The discussion around taxation appears irrelevant to the study findings.

Line 382-383: More detail is required around the budgetary limitation and how it impacted the sample included in the study, which deviates from the protocol previously presented in the article.

Line 385: How was the representitveness of the sample assured?

OVERALL

Consider using people-first language, such as people who smoke instead of smokers. (See: Hefler M, Durkin SJ, Cohen JE, Henriksen L, O'Connor R, Barnoya J, Hill SE, Malone RE. New policy of people-first language to replace 'smoker', 'vaper' 'tobacco user' and other behaviour-based labels. Tob Control. 2023 Mar;32(2):133-134. doi: 10.1136/tc-2023-057950. PMID: 36806099; PMCID: PMC9985717.)

The article would benefit from a careful proofreading to correct grammatical errors.

6. PLOS authors have the option to publish the peer review history of their article (what does this mean? ). If published, this will include your full peer review and any attached files.

**Do you want your identity to be public for this peer review?** For information about this choice, including consent withdrawal, please see our Privacy Policy .

Reviewer #1: No

Reviewer #2: **Yes: ** Harumitsu Suzuki

Reviewer #3: No

---

## [Decision Letter · Decision Letter 1]

13 Dec 2024

PGPH-D-24-01748R1

Examining the Flavor Descriptors of E-Cigarettes, Heated Tobacco Products, and Nicotine Pouches in the Philippines: Regulatory Challenges and Opportunities

Dear Dr. Ackary,

Thank you for submitting your manuscript to PLOS Global Public Health. After careful consideration, we feel that it has merit but does not fully meet PLOS Global Public Health’s publication criteria as it currently stands. Therefore, we invite you to submit a revised version of the manuscript that addresses the points raised during the review process.

We look forward to receiving your revised manuscript.

Kind regards,

Chandrashekhar T. Sreeramareddy

Academic Editor

Journal Requirements:

Additional Editor Comments (if provided):

Dear Authors team,

The manuscript has been re-evaluated by the three reviewers who provided feedback on the initial submission. There are some outstanding concerns specifically on methods sections raised by reviewer 3. please ensure that all the outstanding concerns are addressed in the r2 version.

Reviewers' comments:

Reviewer's Responses to Questions

**Comments to the Author**

1. If the authors have adequately addressed your comments raised in a previous round of review and you feel that this manuscript is now acceptable for publication, you may indicate that here to bypass the “Comments to the Author” section, enter your conflict of interest statement in the “Confidential to Editor” section, and submit your "Accept" recommendation.

Reviewer #1: (No Response)

Reviewer #2: All comments have been addressed

Reviewer #3: (No Response)

2. Does this manuscript meet PLOS Global Public Health’s publication criteria ? Is the manuscript technically sound, and do the data support the conclusions? The manuscript must describe methodologically and ethically rigorous research with conclusions that are appropriately drawn based on the data presented.

Reviewer #1: Yes

Reviewer #2: Yes

Reviewer #3: Partly

3. Has the statistical analysis been performed appropriately and rigorously?

Reviewer #1: Yes

Reviewer #2: Yes

Reviewer #3: N/A

4. Have the authors made all data underlying the findings in their manuscript fully available (please refer to the Data Availability Statement at the start of the manuscript PDF file)?

Reviewer #1: Yes

Reviewer #2: Yes

Reviewer #3: No

5. Is the manuscript presented in an intelligible fashion and written in standard English?

Reviewer #1: Yes

Reviewer #2: Yes

Reviewer #3: Yes

6. Review Comments to the Author

Reviewer #1: The article addresses a relevant public health issue, considering the growing use of ENDS and ENNDS, especially among young people. However, I have three general comments that I believe could improve the current draft:

1. Terminology consistency: The alternating use of "e-cigarettes" and "vapor products" is confusing, particularly since "e-cigarettes" is sometimes used to refer to all the products studied (which include HTP and nicotine pouches); e.g., lines 232 or 373-374. I believe it is standard practice to use "e-cigarettes," though it could be explained upon first mention that they are also called "vapes." The reverse approach could also work, but a single term should be used consistently throughout the text and tables, avoiding the use of "e-cigarettes" when referring to the entire sample of products.

2. Sample size and table totals: It is important to verify that the totals in the tables align with the descriptions in the text. For example:

o Line 218 mentions 280 featured flavor descriptors, but the corresponding column in Table 1 totals 278.

o Line 244 refers to 365 flavor descriptors, yet Table 3 shows 363.

o In Table 2, it is unclear why the total is 46 when the sample should include the country of origin for all products, and no missing data is reported.

3. Sampling Protocol: In lines 154–165, which describe the protocol for fieldwork and product purchases, it is never mentioned that not all unique products available were purchased due to budget constraints. This should be clarified in the Methods section and further elaborated on in the Limitations section (line 559), explaining how many products might have been excluded and the potential implications of this.

Minor Comments:

1. Line 65: A word seems to be missing before "perceived addiction."

2. Line 182: Replace "researched" with "researchers"?

3. Line 193: Remind the reader that the inclusion criteria refer to products "with flavor descriptors”. Also, clarify in Table 1 that the column "with flavor descriptors" corresponds to the inclusion criteria.

4. The title of Table 2 could be more specific about what is included.

5. The discussion about complementary policies, such as taxation and plain packaging, is relevant. However, since this is not directly related to the study results (as acknowledged in lines 515-516), please consider removing it from the abstract’s conclusions.

6. The background in some photographs is very large.

Reviewer #2: I was satisfied with the response to my questions. I have no further questions.

Reviewer #3: Thank you for addressing my suggestions. The manuscript has significantly improved, but there are still major concerns regarding the methodology, particularly related to transparency in data collection and the assessment methods. These issues must be addressed to ensure the rigor and accuracy of the study.

First, the methods section presents an idealized protocol, but it’s only in the limitations section that the authors reveal this protocol was not fully followed due to budget constraints and other challenges. This approach is problematic, as it misrepresents the study’s actual procedures and compromises the transparency and accuracy of the methodology. The methods section should reflect what was actually done, not only what was initially planned. Furthermore, since the team was unable to purchase all unique products, more detail is needed on how a “comprehensive sample collection strategy” was achieved. Please clarify if any specific steps were taken to reduce biases in product selection under these constraints. Without this information, it’s unclear how representative the sample can be.

In addition, it remains unclear how many vendors were visited in each neighborhood, and how those numbers were determined. The manuscript state, “data collection in a neighborhood was considered complete if we collected as many unique products as we could that we had not already purchased from a previous vendor.” How did the team determine that they have collected as many unique products as they could without visiting all stores? How was the definition operationalized? This also appears to contradict the statement that “due to budgetary constraints, our team was unable to purchase all unique e-cigarette products from each retailer we visited.”

Regarding assessment, the methods section only mentions examining flavor descriptors and presence of merchandise and promotional strategies. Were flavor imagery, device shape, and compliance with the Vape Law also systematically assessed, or were those findings based on the team's ad-hoc notes? Please specify this in the methods section.

The authors added, “To confirm the flavor descriptor for RY4, we reviewed the official websites of e-cigarette companies and retailers, where RY4 was consistently described as a blend of tobacco, vanilla, and caramel.” Was this process of confirming descriptors on official websites and checking retailer-provided codebooks consistently followed for all flavors? Please include in the methods section to clarify when and how flavor descriptors were validated.

Finally, the authors added information on how the concept descriptor was defined. This should also be included in the methods section.

7. PLOS authors have the option to publish the peer review history of their article (what does this mean? ). If published, this will include your full peer review and any attached files.

**Do you want your identity to be public for this peer review?** For information about this choice, including consent withdrawal, please see our Privacy Policy .

Reviewer #1: No

Reviewer #2: No

Reviewer #3: No

---

## [Editor Report · Decision Letter 2]

22 Jan 2025

Examining the Flavor Descriptors of E-Cigarettes, Heated Tobacco Products, and Nicotine Pouches in the Philippines: Regulatory Challenges and Opportunities

PGPH-D-24-01748R2

Dear Ms Ackary,

We are pleased to inform you that your manuscript 'Examining the Flavor Descriptors of E-Cigarettes, Heated Tobacco Products, and Nicotine Pouches in the Philippines: Regulatory Challenges and Opportunities' has been provisionally accepted for publication in PLOS Global Public Health.

Best regards,

Chandrashekhar T. Sreeramareddy

Academic Editor

The authors have addressed few outstanding comments on the R1 version. In my own assessment, these were clarifications sought from the reviewers about sample of products and terminology used in the manuscript. The authors team gave satisfactory responses and made suitable amendments in the manuscript. This version of the manuscript is now suitable for publication.